# Benchmarking and Analyzing Unsupervised Network Representation Learning and the Illusion of Progress

**Saket Gurukar**\*                                                   *gurukar.1@osu.edu*
*The Ohio State University*

**Priyesh Vijayan**\*                                         *priyesh.vijayan@mail.mcgill.ca*
*McGill University and Mila*

**Srinivasan Parthasarathy**                                  *srini@cse.ohio-state.edu*
*The Ohio State University*

**Balaraman Ravindran**                                            *ravi@cse.iitm.ac.in*
*Indian Institute of Technology, Madras; Robert Bosch Centre for Data Science and AI*

**Aakash Srinivasan**                                          *s.aakash3431@gmail.com*
*University of California, Los Angeles*

**Goonmeet Bajaj**                                                 *bajaj.32@osu.edu*
*The Ohio State University*

**Chen Cai**                                                        *c1cai@ucsd.edu*
*University of California, San Diego*

**Moniba Keymanesh, Saravana Kumar, Pranav Maneriker**        *keymanesh.1@osu.edu,*
*savkumar90@gmail.com, maneriker.1@osu.edu*
*The Ohio State University*

**Anasua Mitra**                                           *anasuamitra2005@gmail.com*
*Indian Institute of Technology, Guwahati*

**Vedang Patel**                                                   *patel.3140@osu.edu*
*The Ohio State University*

**Reviewed on OpenReview:** *https://openreview.net/forum?id=GvF9ktXI1V*

## Abstract

A number of methods have been developed for unsupervised network representation learning – ranging from classical methods based on the graph spectra to recent random walk based methods and from deep learning based methods to matrix factorization based methods. Each new study inevitably seeks to establish the relative superiority of the proposed method over others. The lack of a standard assessment protocol and benchmark suite often leave practitioners wondering if a new idea represents a significant scientific advance. In this work, we articulate a clear and pressing need to systematically and rigorously benchmark such methods. Our overall assessment – a result of a careful benchmarking of 15 methods for unsupervised network representation learning on 16 non-attributed graphs (several with different characteristics) - is that many recently proposed improvements are somewhat of an illusion when assessed through the lens of downstream tasks such as link prediction and node classification. Specifically, we find that several proposed improvements are marginal at best and that aspects of many of these datasets often render such small differences insignificant,

---

\*First two authors contributed equally to this work.

especially when viewed from a rigorous statistical lens. A more detailed analysis of our results identify several new insights: first, we find that classical methods, often dismissed or not considered by recent efforts, can compete on certain types of datasets if they are tuned appropriately; second, we find that from a qualitative standpoint, a couple of methods based on matrix factorization offer a small but not always consistent advantage over alternative methods; third, no single method completely outperforms other embedding methods on both node classification and link prediction tasks. Finally, we also present several analysis that reveals settings under which certain algorithms perform well (e.g., the role of neighborhood context and dataset properties that impact performance). An important outcome of this study is the benchmark and evaluation protocol, which practitioners may find useful for future research in this area.

# 1 Introduction

Graphs are effective in multiple disparate domains to model, query, and mine relational data. Examples abound, ranging from the use of nearest neighbor graphs in machine learning (Bhattacharyya & Kalita, 2013; Nickel et al., 2016) and database systems (Eppstein et al., 1997; Zhao & Saligrama, 2009) to the analysis of biological networks (Benson et al., 2016; Yue et al., 2019) and from social network analysis (Gu et al., 2017; Zhang et al., 2017) to the analysis of transportation networks (Cetinkaya et al., 2015). Graph or network representation learning seeks to realize a vector representation of each node in the graph (or the graph as a whole) for use in downstream applications such as graph-based recommendation systems (Ying et al., 2018a; Wang et al., 2018), learning molecular fingerprints for drug discovery and cheminformatics (Duvenaud et al., 2015; Lusci et al., 2013), knowledge graphs (Wang et al., 2014; Zhang et al., 2019), anomaly detection (Zhao & Saligrama, 2009; Liang et al., 2018b) and entity resolution (Cohen & Richman, 2002; Getoor & Machanavajjhala, 2012). Fundamental to such applications is the use of classification or predictive methodology based on the underlying representation.

Given the potential applications, a plethora of new network representation learning methods have been proposed recently (See Cai et al. (2018); Hamilton et al. (2017b) for a comprehensive survey). Given the number of such methods, it is difficult for a practitioner to determine or understand which of these methods they should consider adopting for a particular task on a particular dataset. The lack of a standard assessment methodology, makes it difficult to also understand if newer methods truly reflect a scientific advance or if it is simply an illusion of progress (Hand, 2006). Developing a good assessment methodology is both important and daunting.

**Lack of a Standard Evaluation Protocol:** First, the efficacy among embedding methods is often evaluated based on downstream machine learning tasks such as node classification and link prediction. There is no standard evaluation protocol involving a common set of datasets to evaluate the performance on such tasks, which can lead to inconsistent and inconclusive reporting of results. For example, logistic regression is commonly used to evaluate node embeddings' quality for classification, but grid-search over logistic regression parameters is rarely conducted or reported. In our experiments on Blogcatalog, we find that with a train-test split of 50:50 the Laplacian Eigenmaps method (Belkin & Niyogi, 2003) without Grid-Search achieves a Macro-f1 score of **3.9%** (similar to what was reported in (Goyal & Ferrara, 2018; Grover & Leskovec, 2016)). However, by tuning the hyper-parameters, we find that the Laplacian Eigenmaps method achieves a Macro-F1 of **29.2%** (also reported elsewhere (Tang & Liu, 2011)). Similar examples hold for other tasks (e.g. link prediction). Our task setup, preprocessing of datasets and evaluation details are described in Section 6.

**Careful Tuning Of Comparative Methods:** Second, a new method almost always compares its performance against other methods on a subset of tasks and datasets previously evaluated. In many cases, while great care is taken to tune the new method (via careful hyper-parameter tuning) – the same care is often not taken when evaluating baselines. For example, the parameters of Deepwalk, such as the number of walks, walk length, and window size are often set to default or fixed values (Grover & Leskovec, 2016; Cao et al., 2015; Qiu et al., 2018; Tang et al., 2015; Tsitsulin et al., 2018; Wang et al., 2016a; Kipf & Welling, 2016), and

grid search is rarely reported over those Deepwalk parameters (Tang & Liu, 2011). Additionally, reported results are rarely averaged over multiple shuffles to ameliorate bias effects[1]. In short, a lack of consistency in evaluation inhibits our understanding of the scientific advances in this arena.

**Evaluation on Task-Specific Baselines:** Third, for many tasks such as node classification and link prediction, there is a rich pre-existing literature (Bhagat et al., 2011; Lü & Zhou, 2011) focused on such tasks (that do not explicitly rely on node embedding methodology as a preprocessing step). Few, if any, of the prior art in network representation learning consider such baselines – often, such methods compare performance on downstream machine learning tasks against other graph embedding methods.

**Contributions:** To summarize, there is a clear and pressing need for a comprehensive and careful benchmarking of such methods, which is the focus of this study. We perform an experimental study of 15 promising network embeddings methods on 16 diverse non-attributed graphs to address the aforementioned issues in the network embedding literature. The selected embedding methods are unsupervised techniques (many of which have been widely cited) to generate the node embeddings of a graph. Our goal is to perform a uniform, principled comparison of these methods on various datasets and across the two most commonly evaluated tasks – link prediction and node classification.

1. Our most important observation is that progress in this area is somewhat of an illusion (viewed from a rigorous statistical lens), especially since the advent of DeepWalk (Perozzi et al., 2014). Specifically, on the node classification task, *not a single method surveyed offers a statistically significant advantage over DeepWalk*. We find that a couple of methods offer a slight but statistically significant improvement for the link prediction task. In contrast, none of the other methods surveyed offers an advantage over DeepWalk.

2. The task-specific baselines we create are simple and efficient. On the link prediction task, these baselines are competitive with many recent methods (outperformed only by MNMF (Wang et al., 2017) and WYS (Abu-El-Haija et al., 2018)). The task-specific baseline for node classification is competitive on certain types of datasets (datasets with few labels). Other recent methods perform worse on both the tasks than these baselines on a few datasets, suggesting that future researchers on these tasks should consider these task-specific baselines.

3. We present several drill-down analyses that reveal settings under which specific algorithms perform well (e.g., the role of neighborhood context on performance; dataset characteristics that influence performance). We also examine two common ways in which link prediction strategies are evaluated (explicitly through a classifier, or implicitly through vector dot-product ranking). We find that the former is always preferred.

4. An important outcome of this study is the benchmark and evaluation protocol, which practitioners may find useful for future research in this area.

## 2 Notation & Problem Definition

We denote the input graph as $\mathcal{G} = (V, E)$ where $V$ and $E$ denote the set of nodes and edges of the graph, $\mathcal{G}$. The notations used in this work are listed in Table 1. In this study, we consider both directed as well as undirected graphs along with weighted as well as unweighted graphs. We evaluate the embedding methods on non-attributed, homogeneous graphs.

**Network Embedding:** Given a Graph, $\mathcal{G} = (V, E)$ and an embedding dimension, $d$ where $d \ll |V|$, the goal of a network embedding method is to learn a $d$-dimensional representation of each vertex $(V)$ of the graph, $\mathcal{G}$ such that similarity between vertices in graph space approximates to closeness between them in $d$-dimensional space.

| Symbol | Description |
|---|---|
| $A, D, I$ | Adjacency, Degree and Identity Matrix |
| $v_i, w, T$ | Node, Window size, Context window |
| $\Phi(u), \psi(u)$ | Node and Context embedding of node $u$ |
| $U, V$ | Node and context embedding matrix |
| $l_u, N_u, |N_u|$ | Label, neighbor and number of neighbors of node $u$ |
| $\mathbb{1}$ | Indicator function |
| $vol(G)$ | Sum of weights of all edges |
| $b, \lambda$ | Number of negative samples |
| $S$ | Graph Similarity matrix |
| $\sigma(x)$ | Sigmoid function |
| $H$ | Binary community membership matrix |
| $C$ | Latent representations of communities |
| $B$ | Modularity matrix |
| $P$ | $D^{-1}A$ |
| $L$ | Laplacian $L = D - A$ |
| $\mathcal{L}_{1st}, \mathcal{L}_{2nd}$ | Loss functions to preserve first-order and second-order proximities |
| GCN | Graph Convolutional Network |
| $X$ | Node features |
| $q$ | Attention parameter vector |
| $E[D; q]$ | Expectation of the random walk matrix |
| $NS$ | Negative Sampling Distribution. |
| $N, M$ | Number of nodes and negative samples, respectively |
| $\tilde{A}$ | Alternate graph as defined in DGI Velickovic et al. (2019) |
| $X, \tilde{X}$ | Features of Graph $A$ and Alternate Graph $\tilde{A}$ |
| $s$ | Summary vector of Graph. |
| $h_i, \tilde{h}_i$ | Hidden representation of the node $i$ in original and alternate graph |

Table 1: The notations table.

## 3 Network Embedding methods

The methods considered in our study are shown in Table 2. These represent a broad spectrum of classic, popular and recent methods. Due to space limitation, we only present the objective functions of the methods. The notations present in the objective function are defined in Table 1. Interested readers are encouraged to refer to (Belkin & Niyogi, 2003; Perozzi et al., 2014; Grover & Leskovec, 2016; Cao et al., 2015; Qiu et al., 2018; Wang et al., 2017; Ou et al., 2016; Tang et al., 2015; Tsitsulin et al., 2018; Wang et al., 2016a; Kipf & Welling, 2016; Abu-El-Haija et al., 2018; Hamilton et al., 2017a; Velickovic et al., 2019; Huang et al., 2021), for the details about the embedding methods. In the experimental comparison of embedding methods, it becomes critical to reproduce the experimental results. A short summary of the embedding methods are shared in the appendix Section B. To facilitate reproducibility we note the grid search parameters for each method and provide the source code of our evaluation scripts at `https://github.com/PriyeshV/NRL_Benchmark`.

A common parameter for all the embedding methods is the embedding dimension. In this study, we vary the embedding dimensions as [64, 128, 256] and report the best result for the downstream task. On datasets with <300 nodes, we limit the embedding dimensions to 64 and 128. The method-specific parameter values for grid-search are shared in the Table 2.

There are several interesting network embedding methods that utilize node features (Hassani & Khasahmadi, 2020; Peng et al., 2020; Mo et al., 2022; Liu et al., 2022). A survey of such methods including the use of self-supervised learning is presented in (Liu et al., 2022). However, note that unsupervised network representation methods often operate on non-attributed graphs (Leskovec, 2022; Hamilton, 2020) and therefore the methods in Liu et al. (Liu et al., 2022) falls out of the scope of our evaluation. Similarly, other interesting network

---

[1]This is our observation based on the evaluation scripts publically shared by multiple authors.

| Method | Objective function | Reproducibility Notes |
|---|---|---|
| **Manifold-based method** | | |
| Laplacian Eigemaps

(Belkin & Niyogi, 2003) | $\underset{U}{\text{minimize}} \quad trace(U^T L U)$
$\text{subject to} \quad U^T D U = I$ | On the datasets with >1M nodes, Laplacian Eigenmaps did not scale for embedding dimension 128, 256. |
| **Random walk based methods** | | |
| DeepWalk



(Perozzi et al., 2014) | $minimize_\Phi \; -\log Pr(\{v_{i-w}, \cdots, v_{i+w}\} \backslash v_i \mid \Phi(v_i))$ | Walk length = [5, 20, 40], Number of walks = [20, 40, 80], Window size = [2, 4, 10]. In the case of directed graphs, we treat directed graphs as undirected for DeepWalk for improved performance. |
| Node2Vec



(Grover & Leskovec, 2016) | $minimize_\Phi \; -\log Pr(\{v_{i-w}, \cdots, v_{i+w}\} \backslash v_i \mid \Phi(v_i))$ | Walk length = [5, 20, 40], Number of walks = 80. Window Size = 10, $p$ and $q$ = [0.25, 1, 2, 4]. In the case of directed graphs, we treat directed graphs as undirected for Node2Vec for improved performance. |
| **Matrix factorization based methods** | | |
| GraRep



(Cao et al., 2015) | $L_k(u,v) = A_{u,v}^k \cdot \log \sigma(\Phi(u) \cdot \Phi(v))$
$+ \dfrac{\lambda}{|V|} \sum_{v' \in V, v' \neq v} A_{u,v'}^k \cdot \log \sigma(-\Phi(u) \cdot \Phi(v'))$ | $k$ from 1 to 6. On the datasets with >2M edges, due to scalability issue, we searched for $k$ from 1 to 2 and Embedding dimension = [64, 128]. |
| NetMF


(Qiu et al., 2018) | $\log\left(\text{vol}(G)\left(\dfrac{1}{T}\sum_{r=1}^T \left(D^{-1}A\right)^r\right) D^{-1}\right) - \log b$ | $T$ = [1, 10], Negative samples $\lambda$ = [1, 2, 3], Rank $H$ for large context window = [128, 256, 512] |
| MNMF

(Wang et al., 2017) | $O = \underset{U,V,C,H \geq 0}{\min} \|S - VU^T\|^2 + \alpha * \|H - UC^T\|_F^2$
$- \beta tr(H^T B H) + \zeta \|HH^T - I\|_F^2$ | $\alpha$ = [0.1, 1.0, 10.0], $\beta$ = [0.1, 1.0, 10.0] |
| HOPE (Ou et al., 2016) | $min \; \|S - U^s(U^t)^T\|_2^2$ | The decay parameter $\beta = 0.5/\alpha$. $\alpha$ is set to the spectral radius of the graph. We use the weighted Katz as the similarity measure. |
| **Edge reconstruction based method** | | |
| LINE (Tang et al., 2015) | $minimize \sum_{(u,v) \in E} A_{u,v} \; log \; \sigma(\Phi(u).\Phi(v))$
$+ \sum_{(u,v) \in E} A_{u,v} \; log \; \dfrac{exp(\Phi(u).\psi(v))}{\sum_{v' \in V, v' \neq v} exp(\Phi(u).\psi(v'))}$ | Number of samples = 10 billion. In the case of directed graphs, as suggested by the authors of LINE, we evaluate only second-order proximity. |
| **Node similarity based method** | | |
| Verse

(Tsitsulin et al., 2018) | $\sum_{v \in V} \; KL(sim_G(v,.)\|sim_E(v,.))$ | PageRank damping factor $\alpha$ = [0.7, 0.85, 0.9], Negative samples = [3, 10]. |
| **Deep neural network based methods** | | |
| SDNE
(Wang et al., 2016a) | $\mathcal{L}_{joint} = \alpha\mathcal{L}_{1st} + \mathcal{L}_{2nd} + \nu\mathcal{L}_{reg}$ | $\alpha$ = [1e-05, 0.2], Penalty coefficient $\beta$ = [5, 10] |
| VAG
(Kipf & Welling, 2016) | $\hat{A} = \sigma(ZZ^T); \; Z = GCN(X, A)$ | Hidden layers size = [128, 256] |
| Watch Your Step

(Abu-El-Haija et al., 2018) | $\underset{R,S,q}{min} \quad \beta\|q\|_2^2 - \|E[D;q] \circ \log(\sigma(U*V^T))$
$- \mathbb{1}[A=0] \circ \log(1 - \sigma(U*V^T))\|_1$ | Number of Hops = 5 |
| **Table continued on the next page...** | | |

| Method | Objective function | Reproducibility Notes |
|---|---|---|
| **Table continued from the previous page...** | | |
| Unsupervised-GraphSage

(Unsup_GS) (Hamilton et al., 2017a) | $-log(\sigma(\phi(u)^T\phi(v)) - \lambda. \mathbb{E}_{w \sim NS} \, log(\sigma(-\phi(u)^T\phi(w))$ | Walk length = [5, 20, 40], Number of walks = [20, 40, 80] |
| DGI (Velickovic et al., 2019) | $\frac{1}{N+M} \left( \sum_{i=1}^{N} \mathbb{E}_{(X,A)} \left[ log \, D(h_i, s) \right] \right.$ $\left. + \sum_{j=1}^{M} \mathbb{E}_{(\tilde{X}, \tilde{A})} \left[ log \, (1 - D(\tilde{h_j}, s)] \right) \right)$ | Number of layers=[1, 2, 3], Hidden layer size=[64,28] and with and without freezing gradient flow for I/p embedding and GCN for the negative network |
| RW-framework (Huang et al., 2021) | Factorization or Sampling | Markov_time = [3, 50, 100] Similarity = ["autocovariance", "PMI"] Algorithm = ["factorization", "sampling"] |
| Poincaré (Huang et al., 2021) | $\mathcal{L}(\theta) = \sum_{(u,v) \in E} log \frac{e^{-d(u,v)}}{\sum_{v' \in N'_v} e^{-d(u,v')}}$ $d(u,v) = arccosh \left( 1 + 2 \frac{\|u-v\|^2}{(1-\|u\|^2)(1-\|v\|^2)} \right)$ | Number of negatives = [10, 30, 50, 70, 90], Manifolds= [Poincare, Lorentz], Models=[Distance, Energy] |

Table 2: The embedding methods evaluated in our study. See Table 1 for objective function notation key. The following code bases were in Python: Laplacian Eigenmaps, DeepWalk, GraRep, NetMF, MNMF, HOPE, SDNE, VAG, WYS, Unsupervised GraphSage, DGI and RW-framework. The following code bases were in C or C++: Node2Vec, Verse, LINE. All are widely used code bases typically from the original authors. For methods that leverage a learning rate parameter, we varied across following rates [ 0.1, 0.5, 1.0]. For methods that lever epochs parameter, we varied across following number of epoch values [ 50, 100].

embedding methods (Hassani & Khasahmadi, 2020; Peng et al., 2020; Mo et al., 2022) that require node attributes are not considered in our evaluation.

Alternatively there are some interesting network embedding methods that focuses on imposing constraints on the embedding space (Nickel & Kiela, 2017; 2018) and do not require node attributes. For instance, Nickel et al. (Nickel & Kiela, 2017; 2018) propose learning the node embeddings in the hyperbolic space instead of euclidean space. We evaluate and include such methods in our empirical analysis.

Other machine learning downstream tasks such as learning fair node embeddings (Agarwal et al., 2021) or performing graph classification Togninalli et al. (2019); Ying et al. (2018b) have recently received significant attention. For instance, Agarwal et. al (Agarwal et al., 2021) learns node embeddings in an efficient and fair manner. However such methods require sensitive attributes to both be present and pre-identified and is therefore not applicable to our setting. Other works Togninalli et al. (2019); Ying et al. (2018b) focus on learning the graph representations for tasks such as graph classification. However, in this work, we mainly focus on two downstream tasks: node classification and link prediction tasks. The generic prepossessing of adjacency matrix has also been explored (Klicpera et al., 2019). GCC (Klicpera et al., 2019) introduces a generic two-step pre-processing that recommends the usage of a sparsified higher-order transition matrix (Page rank and Heat diffusion) instead of the original adjacency matrix. Such methods rely on substituting a simple transition matrix with a higher-order transition matrix for graph convolutions in GNNs. We do not include GCC method in our evaluation as it is not a standalone embedding method and focuses on preprocessing the adjacency matrix.

# 4   Related Work & Feedback

**Related Work:** Network representational learning has attracted a lot of attention in the past few years. Recent surveys (Cai et al., 2018; Hamilton et al., 2017b; Zhang et al., 2018a) attempt to categorize or taxonomize these efforts along various axes but they do not offer an empirical analysis of the embedding methods. Slightly prior to our study, we were aware of one other excellent empirical study of network embedding methods (Goyal & Ferrara, 2018). However, there are significant differences between this effort and ours. First, our systematic study encompasses over 375,000 individual experiments, on a larger set of embedding methods, including several more recent ideas and on many more datasets (16 vs 7) than the

previous study. Specifically, we evaluate 15 embedding methods + 2 efficient heuristics on 16 datasets. Second, we carefully tune all hyperparameters of each method as well as the logistic classifier (and include this information in our reproducibility notes). As a concrete example of where such careful tuning can make a difference consider that on Blogcatalog with a train-test split of 50:50, Goyal *et al.* (Goyal & Ferrara, 2018), achieve a Macro-f1 score of 3.9% while with tuning the hyper-parameters of logistic regression we achieve a Macro-f1 score of 29.2%. Third, our analysis reveals several important insights on the role of context, the role of different link prediction evaluation strategies (dot product vs classifier), the impact of non-linear classifiers, and many others. All of these provide useful insights for end-users as well as guidance for future research and evaluation in network representational learning and downstream applications. Fourth, we also provide a comparison against simple but effective task-specific baseline heuristics which will serve as useful strawman methods for future work. Finally and perhaps most distinctively, we carefully analyze comparative performance through a careful grid search and rigorous statistical lens, pointing out that many recent advances are perhaps somewhat of an illusion – this is not a conclusion drawn from the previous effort.

Subsequent to our initial work, we also learned of another effort by Khosla et al. (Khosla et al., 2019), who conducted an empirical bake-off of unsupervised network embedding methods and found that no single method always outperformed other methods – a consistent message with both our work and that of Goyal et al. (Goyal & Ferrara, 2018). The differences between our work and this work can be similarly outlined as above (comparative scale of experiments on a wider range of methods and datasets, detailed hyper-parameter tuning and comparison with task-specific heuristics). Again and perhaps most distinctively, the authors do not analyze comparative performance through a rigorous statistical lens or attempt to examine if recent research represents a true advance or an illusion of progress. Both sets of authors also do not: i) study the impact of linear vs non-linear classifier on node classification task; ii) the difference between dot-product and classifier for link prediction task; and iii) the impact of embedding dimension/embedding normalization on both node classification and link prediction tasks.

**Community Feedback:** After our initial work, we socialized and sought feedback within the database, data science and machine learning communities. We received several useful feedback on this work as follows.

- It was suggested that we should include an unsupervised variant of Deep Graph Infomax (Velickovic et al., 2019) in our study. We have added the experiments with DGI.

- It was suggested that we should compare with some Graph Neural Network methods (Kipf & Welling, 2017). We note that Graph Neural Networks typically fall under the "semi-supervised learning" paradigm. However, we do perform experiments on unsupervised variants of Graph Convolutional Network proposed by Kipf & Welling (2016) and Velickovic et al. (2019).

- Finally we received several suggestions for additional experiments (see main results and appendix), evaluation on new datasets (e.g. Reddit), and positioning that has helped shape both the analysis and eventual presentation of results.

## 5   Datasets

Table 3 describes empirical properties of the datasets in our study. The datasets support both standard multi-class classification (each node is associated a single class label from among multiple class labels) as well as multi-label classification (a node can be associated with multiple class labels). Directed as well as undirected datasets are considered in order to evaluate the embedding methods on the link-prediction task efficiently. Further, datasets with and without edge weights are also included. The datasets exhibit varying levels of homogeneity where homogeneity of a node (say $u$) is computed using the Jaccard function as : $HG(u) = \sum_{i \in N_u}(Jaccard(l_u, l_i))/|N_u|$, where $N_u$ represents the neighborhood set of $u$. Note this formulation naturally handles both multi-class[2] and multi-label datasets. The homogeneity value of a dataset is represented by the average homogeneity value of nodes in the dataset. Note that for the publicly available YouTube dataset (Zafarani & Liu, 2009), we have node classes for only 2.7% nodes; hence we do not report

---

[2]For the multi-class setting this is equivalent to $HG(u) = \sum_{i \in N_u} \mathbb{1}(l_u == l_i)/|N_u|$ used elsewhere (Ma et al., 2021).

| Dataset | #N | #E | #L | (C/L) | D | W | CC | HG | SP |
|---|---|---|---|---|---|---|---|---|---|
| WebKB (Texas)  (Craven et al., 1998) | 186 | 464 | 4 | C | F | T | 0.195 | 0.405 | 2.70E-02 |
| WebKB (Cornell)  (Craven et al., 1998) | 195 | 478 | 5 | C | F | T | 0.157 | 0.436 | 2.53E-02 |
| WebKB (Washington)  (Craven et al., 1998) | 230 | 596 | 5 | C | F | T | 0.197 | 0.470 | 2.26E-02 |
| WebKB (Wisconsin)  (Craven et al., 1998) | 265 | 724 | 5 | C | F | T | 0.208 | 0.437 | 2.07E-02 |
| PPI  (Breitkreutz et al., 2007) | 3,890 | 38,739 | 50 | L | F | F | 0.242 | 0.109 | 5.12E-03 |
| Wikipedia  (Mahoney, 2011) | 4,777 | 92,517 | 40 | L | F | T | 0.569 | 0.116 | 8.11E-03 |
| Blogcatalog  (Zafarani & Liu, 2009) | 10,312 | 333,983 | 39 | L | F | F | 0.463 | 0.145 | 6.28E-03 |
| DBLP (Co-Author) (Tang et al., 2008) | 18,721 | 122,245 | 3 | C | F | T | 0.833 | 0.453 | 6.98E-04 |
| CoCit (Microsoft)  (Tsitsulin et al., 2018) | 44,034 | 195,361 | 15 | C | F | F | 0.142 | 0.423 | 2.02E-04 |
| Wiki-Vote  (Leskovec et al., 2010) | 7,115 | 103,689 | - | - | T | F | 0.141 | - | 4.10E-03 |
| Pubmed (Namata et al., 2012) | 19,717 | 44,338 | 3 | C | T | F | 0.060 | 0.792 | 2.28E-04 |
| p2p-Gnutella  (Ripeanu & Foster, 2002) | 62,586 | 147,892 | - | - | T | F | 0.005 | - | 7.55E-05 |
| Flickr  (Zafarani & Liu, 2009) | 80,513 | 5,899,882 | 195 | L | F | F | 0.165 | 0.146 | 1.82E-03 |
| Epinions (Richardson et al., 2003) | 75,879 | 508,837 | - | - | T | F | 0.137 | - | 1.77E-04 |
| YouTube (Zafarani & Liu, 2009) | 1,134,890 | 2,987,624 | 47 | C | F | F | 0.081 | - | 4.64E-06 |
| Reddit (Hamilton et al., 2017a) | 231,443 | 11,606,919 | 41 | C | F | F | 0.169 | 0.818 | 4.33E-04 |

Table 3: Dataset Properties. Notations: Nodes(N), Edges(E), Labels (L), Multi-Class (C)/ Multi-Label (L), Directed (D) and Weighted (W). Clustering coefficient (CC), Homogeneity (HG), and Sparsity (SP)

its homogeneity score in Table 3. For each dataset we also report its clustering coefficient - a commonly used measure of local density(Newman, 2010). We also report sparsity for each dataset, defined as fraction of actual edges over number of maximum possible edges. The datasets span various domains: Web (WebKB), Medical (PPI), Natural Language (Wikipedia), Social (Blogcatalog, Flickr, YouTube, Epinions), Citation (DBLP, CoCit, and Pubmed), Digital (p2p-Gnutella) and Voting (Wiki-vote). The datasets Wiki-vote, p2p-Gnutella, and Epinions do not contain node labels. Hence, on those datasets, we evaluate the embeddings methods only on the link prediction task.

We next present the list of datasets and their prior use in evaluating unsupervised network representation learning methods: WebKB datasets (Wang et al., 2017; Mitra et al., 2020; Li et al., 2018; Jin et al., 2018; Adhikari et al., 2018; Al-Sayouri et al., 2020; Wang et al., 2016b), PPI (Grover & Leskovec, 2016; Qiu et al., 2018; Abu-El-Haija et al., 2018; Alsentzer et al., 2020; Mitra et al., 2020; Epasto & Perozzi, 2019), Wikipedia (Grover & Leskovec, 2016; Qiu et al., 2018; Tang et al., 2015; Al-Sayouri et al., 2020; Mitra et al., 2020; Cavallari et al., 2017), Blogcatalog (Perozzi et al., 2014; Grover & Leskovec, 2016; Cao et al., 2015; Qiu et al., 2018; Tang et al., 2015; Tsitsulin et al., 2018; Wang et al., 2016a; Mitra et al., 2020; Qiu et al., 2018; Cavallari et al., 2017; Li et al., 2017), DBLP (Tsitsulin et al., 2018; Cao et al., 2015; Mitra et al., 2020; Cavallari et al., 2017; Li et al., 2017), Microsoft (Tsitsulin et al., 2018; Mitra et al., 2020; Wang et al., 2020; Spitz et al., 2018), Wiki-Vote (Abu-El-Haija et al., 2018; Agrawal et al., 2013; Heidari et al., 2015; Epasto & Perozzi, 2019; Sun et al., 2019), Pubmed (Kipf & Welling, 2016; Velickovic et al., 2019; Mitra et al., 2020; Wang et al., 2020), Flickr  (Perozzi et al., 2014; Grover & Leskovec, 2016; Qiu et al., 2018; Tang et al., 2015; Wang et al., 2016a; Li et al., 2018; Cavallari et al., 2017; Li et al., 2017), P2P-Gnutella  (Tu et al., 2019b; Kim et al., 2018; Zhao et al., 2013; Moon et al., 2020), Epinions  (Richardson et al., 2003; Agrawal et al., 2013; Epasto & Perozzi, 2019; Li et al., 2017), Reddit  (Hamilton et al., 2017a; Chiang et al., 2019; Zhang et al., 2020; Liu et al., 2020; Velickovic et al., 2019), and YouTube  (Perozzi et al., 2014; Tang et al., 2015; Tsitsulin et al., 2018; Wang et al., 2016a). While each dataset has been evaluated on several methods, there is not a single dataset common to all these efforts - making it difficult to compare different network embedding methods meaningfully. Note that the focus of the the unsupervised network representation learning methods has primarily been on preserving the structure. Hence, these methods often operate on non-attributed graphs (Leskovec, 2022; Hamilton, 2020).

## 6  Experimental setup

In this section, we elaborate on the experimental setup for link prediction and node classification tasks employed to evaluate the quality of embeddings generated by different methods. Additionally, we also

present two heuristics based baselines for both the tasks and define the metrics used for comparing the embedding methods.

## 6.1 Link Prediction

Prediction of ties is an essential task in multiple domains where the relational information is costlier to obtain such as drug-target interactions (Crichton et al., 2018), protein-protein interactions (Airoldi et al., 2006), or when the environment is partially observable. "Link prediction, ... consists of inferring the existence of new relationships or still unknown interactions between pairs of entities based on their properties and the currently observed links" (Martínez et al., 2016). In this work, we focus on the prediction of unknown interactions (links) between two nodes. The problem of predicting a tie/link between two nodes $i$ and $j$ is often evaluated in one of two ways. The first is to treat the problem as a binary classification problem. The second is to use the dot product on the embedding space as a scoring function to evaluate the tie's strength.

The edge features for binary classification consists of node embeddings of nodes $i$ and $j$, where two node embeddings are aggregated with a binary function. In our study, we experimented with three binary functions on node embeddings: Concatenation, Hadamard and L2 distance. Node2vec (Grover & Leskovec, 2016) also levered the binary functions Hadamard, and L2 distance. We use Logistic Regression (LR) as our base classifier for the task of link prediction. The parameters of the logistic regression are tuned using a grid search with 5-fold cross-validation. We evaluate the link prediction performance with metrics: Area Under the Receiver Operating Characteristics (AUROC). An alternative evaluation strategy is to predict the presence of link $(i, j)$ based on the dot product value of node embeddings of nodes $i$ and $j$. We study the impact of both the evaluation strategies in Section 7.1.

**Construction of the train and test sets**: The method of construction of train and test sets for link prediction task is crucial for comparing embedding methods. The train and test split consists of 80% and 20% of the edges, respectively, and are constructed in the following order:

1. Self-loops are removed.
2. We randomly select 20% of all edges as positive test edges and add them in the test set.
3. Positive test edges are removed from the graph. We find the largest weakly connected component formed with the non-removed edges. The edges of the connected component form positive train edges.
4. We sample negative edges from the largest weakly connected component (C) and add the sampled negative edges to both the training and test sets where a negative edge is defined as $(i, j) \notin C$. The number of negative edges is equal to the number of positive edges in both training and test sets.
5. For directed graphs, we form "directed negative test edges" which satisfy the following constraint: $(j, i) \notin C$ but $(i, j) \in C$ where $C$ refers to edges in the largest weakly connected component. We add the directed negative test edges $(j, i)$ edges to our test set. The number of "directed negative test edges" is a parameter and in our experiments we set it to 10% of negative test edges in the test set.
6. Nodes present in the test set, but not present in the training set, are deleted from the test set. For learning the binary classifier, all the edges in the training set are considered positive train edges. We sample an equal number of negative train edges from the training set. Edges from the training set as positive train edges and add an equal number of negative train edges

In the case of large datasets (>5M edges), we reduce our training set. We consider 10% of both randomly selected positive and negative train edges for learning the binary classifier. The learned model is evaluated on the test set. The above steps are repeated for 5 train:test splits of 80:20, and we report the average AUROC across 5 splits. The additional details about the folds are shared in appendix Section C.

**Hyperparameter tuning:** As stated previously, our focus is on "unsupervised" network representation learning methods. We perform the grid search over the method parameters and get a score on the test split for all possible parameters in the grid. We report the best score from this search for each method. We note that such a strategy has been previously employed by others (see (Abu-El-Haija et al., 2018; Huang et al.,

2021; Ou et al., 2016; Wang et al., 2017). The hyperparameters of the method are shared in Table 2 under column "Reproducibility Notes".

**Preprocessing procedure**: The preprocessing procedure is visually described in the appendix C and the statistics of train:test splits of dataset is shared in Table 4 and Table 5. In brief, we consider five train:test splits of 80:20 rather than a single split and report the average performance over the five splits. Like all approaches in the literature we require the training graph to be connected. Specifically, following the approach of Goyal & Ferrara (2018) we compute a weakly connected component for identifying the connected training graph Goyal & Ferrara (2018). As an alternative, other methods remove a set of edges from the original graph, while ensuring that the training graph remains connected Huang et al. (2021).

## 6.2 Node classification

In the network embedding literature, node classification is the most popular way of comparing the quality of embeddings generated by different embedding methods. The generated node embeddings are treated as node features, and node labels are considered as ground truth. The classification tasks performed in our experiments are either multi-label or multi-class classification. The details on the classification task performed on each dataset are provided in Table 3. We select Logistic Regression as our classifier. We split the dataset with 50:50 train-test splits. In the case of large datasets (Flickr and YouTube), we split the dataset with 5:95 train-test splits based on labeled nodes. The learned model is evaluated on the test set, and we report the results averaged over 10 shuffles of train-test sets.

We noted that most of the efforts in the literature do not tune the hyper-parameters of Logistic Regression. Default hyper-parameters are not always the best hyper-parameters for Logistic Regression. For instance, with the LR classifier's default hyper-parameters, the Macro-f1 performance of Laplacian eigenmaps on the Blogcatalog dataset is 3.9% for the train-test split of 50:50. However, tuning the hyper-parameters results in significant improvement of the Macro-f1 score to 29.2%. In our experiments, we perform five-fold cross-validation with the following parameters of logistic regression: "inverse of regularization strength" C=[0.001, 0.01, 0.1, 1, 10, 100] and "norm used in the penalization"=['L1' and 'L2']. We lever the logistic regression implementation provided by scikit-learn (Pedregosa et al., 2011). We also explored Bayesian optimization for identifying the best Logistic Regression or EigenPro parameters with the help of auto-sklearn (Feurer et al., 2020). However, even given a time budget of two days, the auto-sklearn results were rarely able to even match (never beat) the results computed with the grid-search suggesting that our grid search was comprehensive.

The choice of a "linear" classifier to evaluate the quality of embeddings is not a hard constraint in the node classification task. In this work, we also test the idea of leveraging a "non-linear" classifier for the node classification task and compare its performance to that of a linear classifier. We use the EigenPro (Ma & Belkin, 2017) classifier as our non-linear classifier. EigenPro (Ma & Belkin, 2017) provides a significant performance boost over the state-of-the-art kernel methods with faster convergence rates on large datasets. In our experiments, we see up to 15% improvement in Micro-f1 scores with non-linear classifiers compared to the linear classifier. (See Section 7.2 for details).

## 6.3 Task-Specific Baselines

Next, we present new baselines for both link prediction and node classification tasks. The purpose of defining such a baseline is to assess the difficulty of different tasks on a different dataset and compare the performance of sophisticated network embedding methods over simple heuristics.

### 6.3.1 Link Prediction Heuristic

In the link prediction literature, there exist multiple similarity-based metrics (Lü & Zhou, 2011), which can predict a score for link formation between two nodes. Examples of such metrics include Jaccard Index (Jaccard, 1901; Wang et al., 2007), Adamic Adar (Adamic & Adar, 2003). These similarity-based metrics often base their predictions on the neighborhood overlap between the nodes. We combine the similarity-based metrics to form a curated feature vector of each edge (Sinha et al., 2018). The binary classifier in the link

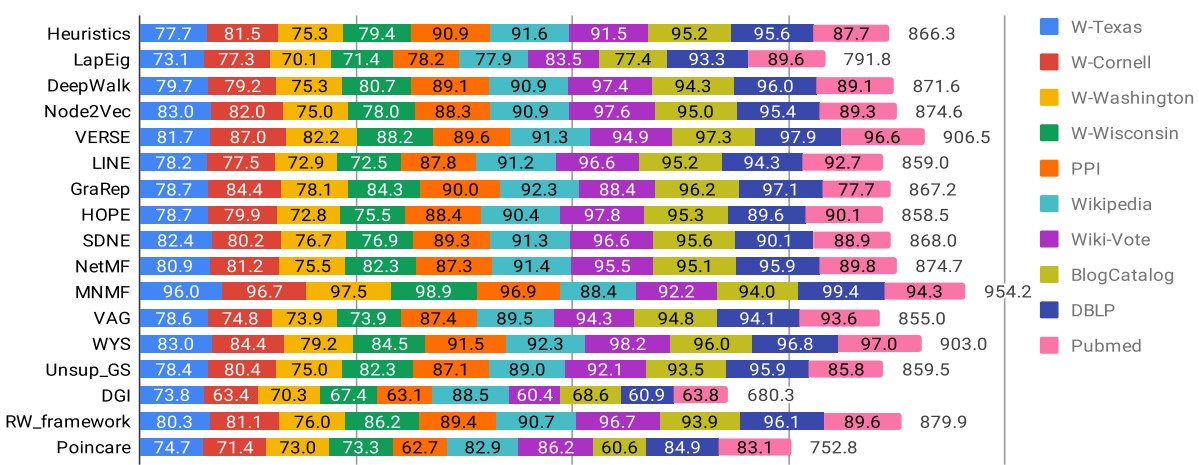

(a) Smaller datasets: All methods complete execution.

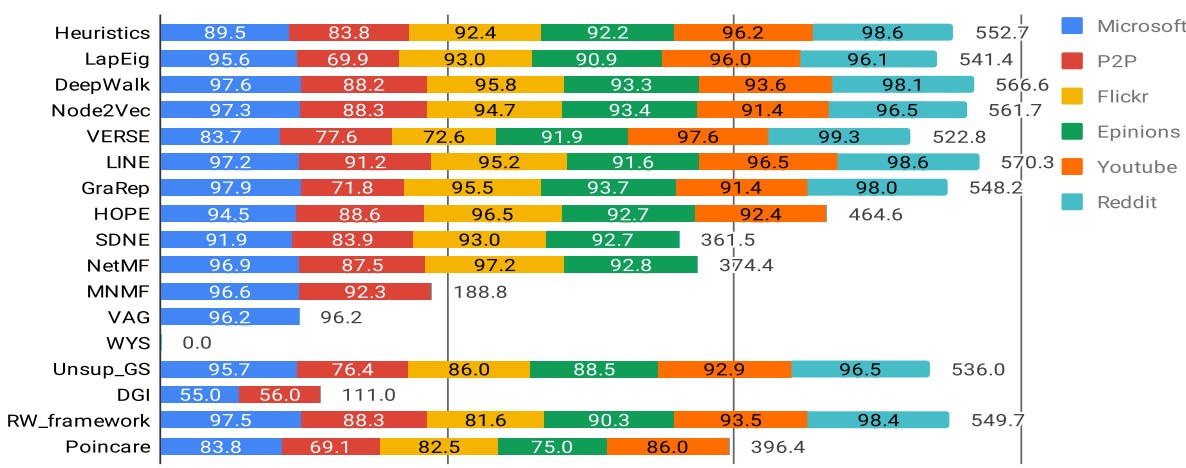

(b) Larger datasets: Not all methods complete execution.

Figure 1: Link Prediction performance measured with **AUROC**. The AUROC score is multiplied by 100 for better readability. Not all methods are scalable on the datasets shown in Figure 16b.

prediction task is then trained on these generated edge features. Our selected similarity-based metrics are Common Neighbors (CN), Adamic Adar (AA) (Adamic & Adar, 2003), Jaccard Index (JA) (Jaccard, 1901), Resource Allocation Index (RA) (Zhou et al., 2009) and Preferential Attachment Index (PA) (Barabási & Albert, 1999). The similarity-based metrics CN, JA, and PA, capture first-order proximity between nodes, while the metrics AA and RA capture second-order proximity between nodes. We found this heuristic-based model to be highly competitive when compared to the embedding methods on multiple datasets.

### 6.3.2 Node Classification Heuristic

Nodes in the graph can be characterized (represented) by their properties. We combine the node properties to form a feature vector (embedding) of a node. The classifier in the node classification task is then trained on this node embedding. The node properties capture information such as nodes' neighborhood, influence on other nodes and their structural properties. We select following node properties: Degree, PageRank (Page

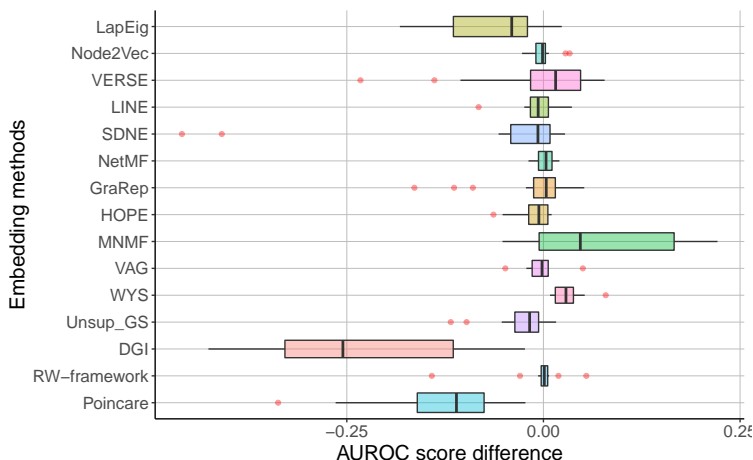

Figure 2: Illusion of Progress I: Link Prediction: Box-plot represents the distribution of differences between AUROC of the embedding method and AUROC of Deepwalk.

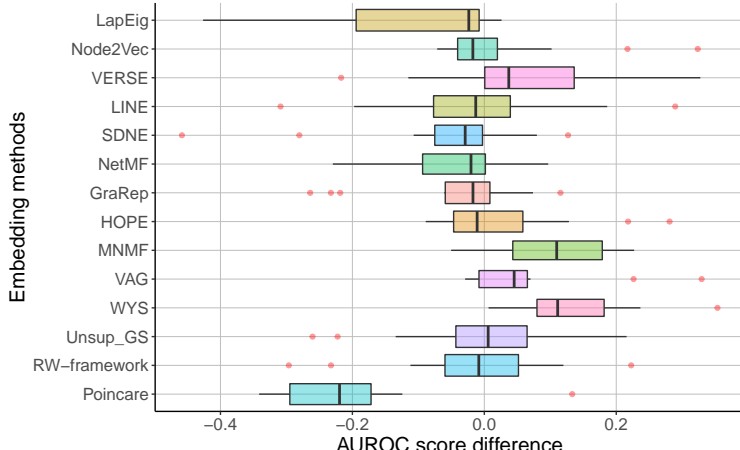

Figure 3: Illusion of Progress I: Link Prediction (dot product). Box-plot represents the distribution of differences between AUROC of the embedding method and AUROC of Deepwalk.

et al., 1999), Clustering Coefficient, Hub and Authority scores (Kleinberg, 1999), Average Neighbors' Degree, and Eccentricity (Newman, 2010). We treat the graph as undirected while computing the node properties. As each node property's magnitude varies with another, we perform column-wise normalization with RobustScaler available from Scikit-learn. We show in the experiments Section 7.2 that the node classification heuristics baseline is competitive with several recent embedding methods on datasets with few (up to five) labels.

# 7 Experimental Results

This section reports the performance of network embedding methods on link prediction and node classification tasks. We tune both the parameters of embedding methods and classifiers' parameters in link prediction and node classification tasks. Whenever possible, we rely on the authors' code implementation of the embedding method. All methods that do not finish for large datasets are executed on a modern machine with 500 GB RAM and 28 cores. All the evaluation scripts are executed in the same virtual python environment.

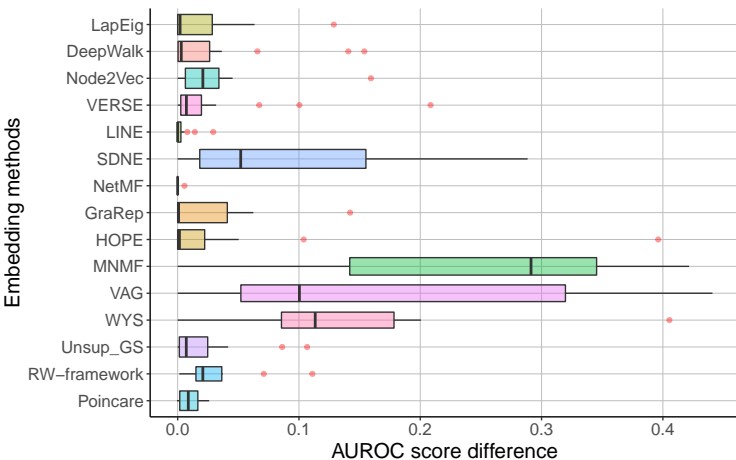

Figure 4: Default vs Tuned Parameters: Link Prediction. Box-plot represents the distribution of differences between AUROC of the embedding method with default parameters vs tuned parameters.

## 7.1 Link Prediction

The link prediction performance of 15 embedding methods measured in terms of AUROC on 16 datasets is shown in Figure 1. The *Overall* (or aggregate) performance of an embedding method on all the datasets is also shown at the end of the horizontal bar of each method in Figure 1. We represent *Overall* score as the sum of scores (AUROC) of the method on all the datasets. As mentioned in Section B, we tune the hyper-parameters of each embedding method and report the best average AUROC scores across 5 train:test splits. We perform the link prediction task with both normalized and unnormalized embeddings and report the best performance. We make the following observations:

**A. Illusion of Progress** : We begin by noting that the current emphasis and interest in network representation learning can partially be attributed to the Deepwalk's Perozzi et al. (2014) simplicity. In this analysis, we seek to understand if more recent advances offer a real advancement (or if it is an illusion) over Deepwalk on the link prediction task. The distribution of these differences with respect to each embedding method versus Deepwalk is shown as box-plot in Figure 2. A positive difference implies that the embedding method's performance is better than Deepwalk in terms of AUROC. The improvement over Deepwalk is statistically significant by paired t-test only for MNMF and WYS (marginal) with a significance level of 0.05. We observe that except for MNMF and WYS the performance of Deepwalk is comparable or superior to the others. We also observe that the median of the box-plot for *link prediction heuristic* is also close to that of Deepwalk. WYS provides marginally better link prediction performance over Deepwalk and the heuristic-based model on the smaller datasets. MNMF is the only method that seems to offer a small but consistent advancement over the state-of-the-art suggesting it is the only counterexample to the assertion that recent advances are largely an illusion at least as it relates to the downstream task of link prediction when evaluated on non-attributed graphs. Similar to the results shown in Figure 2 with classifier, we observe that with dot-product too, the improvement over Deepwalk is statistically significant by paired t-test for only MNMF and WYS with a significance level of 0.05.

**Efficacy of Deepwalk on Link Prediction task**: We observe that the link prediction performance of Deepwalk and other random-walk based methods improves significantly through the use of classifier (see section D). Moreover, Deepwalk is able to effectively capture the local neighborhood of node in the embedding through multiple random walks and the captured neighborhood information appears sufficiently expressive for the link prediction task.

**B. Effectiveness of Link Prediction Heuristic:** The heuristic baseline – described in section 6.3.1 – is very effective while being simple to compute and efficient (not shown as this study focuses entirely on quality). On the largest dataset YouTube, the heuristic achieves an AUROC of 96.2%, close to the best performing

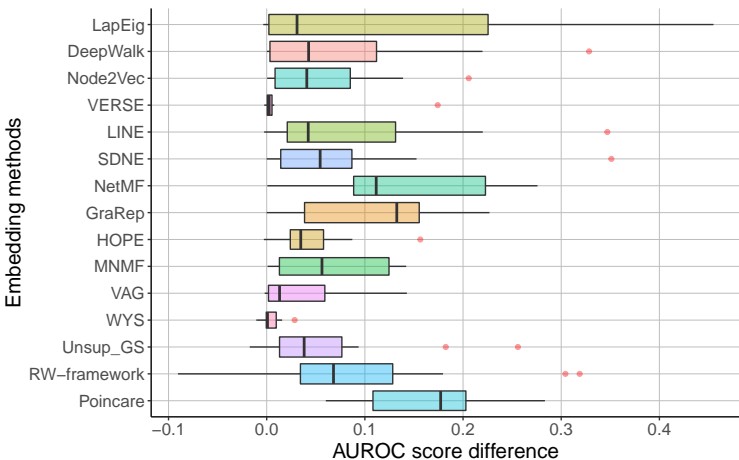

Figure 5: The box-plot represents distribution of the differences between AUROC score with dot product and AUROC score with classifier on all the datasets.

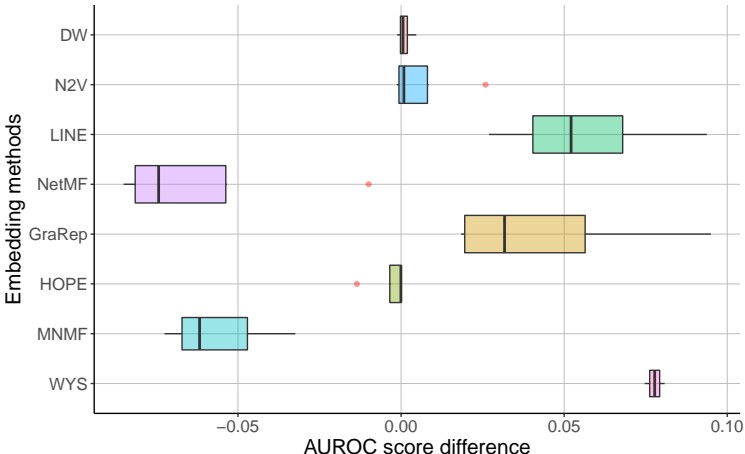

Figure 6: The box-plot represents the distribution of the differences between AUROC score with node + context embeddings and AUROC score with only node embeddings on directed datasets.

Verse model with an AUROC of 97.6%. When compared to the most competitive baseline MNMF, the link prediction heuristic outperforms MNMF on Wikipedia and Blogcatalog datasets. We also observe that the task-specific baseline's performance is competitive against several other methods on the directed datasets despite using undirected similarity-based metrics. The core components of the link prediction heuristic capture important first- and second-order local topological proximity information which are essential for link prediction and may help explain its impressive performance.

**C. Effectiveness of MNMF for Link Prediction**: We observe that MNMF achieves the highest overall link prediction performance in terms of best average AUROC. We also observe that MNMF does not always outperform other methods on all the datasets. For instance, on the Wiki-Vote and Pubmed dataset, WYS achieves the best average AUROC scores while on the Microsoft dataset, GraRep achieves the best average AUROC score. Note that MNMF did not scale for the datasets with $\geq$ 5M edges on a modern machine with 500 GB RAM and 28 cores. However, the scalability issue of non-negative matrix factorization based methods can be addressed by adopting recent ideas Moon et al. (2019); Liang et al. (2018a) (outside the scope of this study). We note that community structure is an important topological property of the network and can play a significant role in predicting links within a network. MNMF by construction, explicitly imposes a higher level of constraint on the node representation, and tries to ensure that the vector representation of

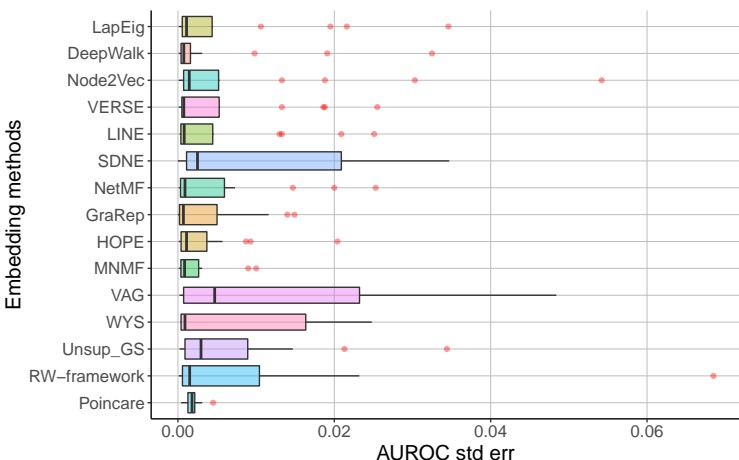

Figure 7: The box-plot represents the distribution of the standard error of AUROC scores computed for each method on each dataset across five splits.

nodes within a community are more similar than nodes across different communities. This may explain its improved performance over peer methods on the link prediction task.

**D. Impact of Evaluation Strategy:** As described in section 6.1, the presence of a link between two nodes can be predicted with either the Logistic Regression classifier (treating the embeddings as features) or the dot product between the node embeddings. The latter is the widely used approach in the literature Ou et al. (2016); Kipf & Welling (2016); Abu-El-Haija et al. (2018); Wang et al. (2016a); Goyal & Ferrara (2018). Here, we compare both the evaluation strategies' performance on each embedding method across all datasets using the average AUROC scores' differences. The results are presented as box-plot in Figure 5. A positive difference implies that the classifier's performance is better than the dot product score in terms of AUROC. The paired t-test suggests the positive difference is statistically significant for all methods, except for Verse, VAG and WYS, with a significance level of 0.05. For Verse and WYS, both strategies resulted in near-identical performance (almost no change). In summary, we find that using a classifier over dot product provides significant predictive performance gain on the link prediction task and probably should be the evaluation strategy of choice moving forward for this task. Note that, irrespective of the evaluation strategy (classifier or dot-product), we observe that only two methods (MNMF and WYS) outperform Deepwalk in a statistical significant manner. Result for link prediction task with dot product is not shown here due paucity of space.

**E. Impact of context embeddings:** We study context embeddings' impact on directed datasets for the link prediction task. We consider only those embedding methods which generate both node and context embeddings for this study. We compare the impact of using node + context embeddings over using only node embeddings with the help of differences in AUROC scores. The "+" symbol refers to the concatenation operation. The results are detailed in Figure 6. A positive difference implies that node + context embeddings' performance is better than just node embeddings in terms of AUROC. The difference is statistically significant (paired t-test) with a significance level 0.05 for LINE and WYS for which using context information helps. In contrast, for GraRep and MNMF, it hurts the performance. Context also appears to help HOPE, but it is not statistically significant. We see that levering node + context embeddings improve the link prediction performance of LINE, HOPE, and WYS. For MNMF, context embeddings do not improve the link prediction performance because, in MNMF, the community information – crucial for link prediction – is already incorporated in the node embeddings. In the case of GraRep, we find that the node embeddings encapsulate high-order information and, hence, levering context does not help improve the performance.

**F. Stability and Robustness:** For link prediction, we thus far report the average AUROC score of an embedding method over 5 splits of a selected dataset. The computed average AUROC standard error corresponds to a measure of the robustness (or stability) of that embedding method on the selected dataset

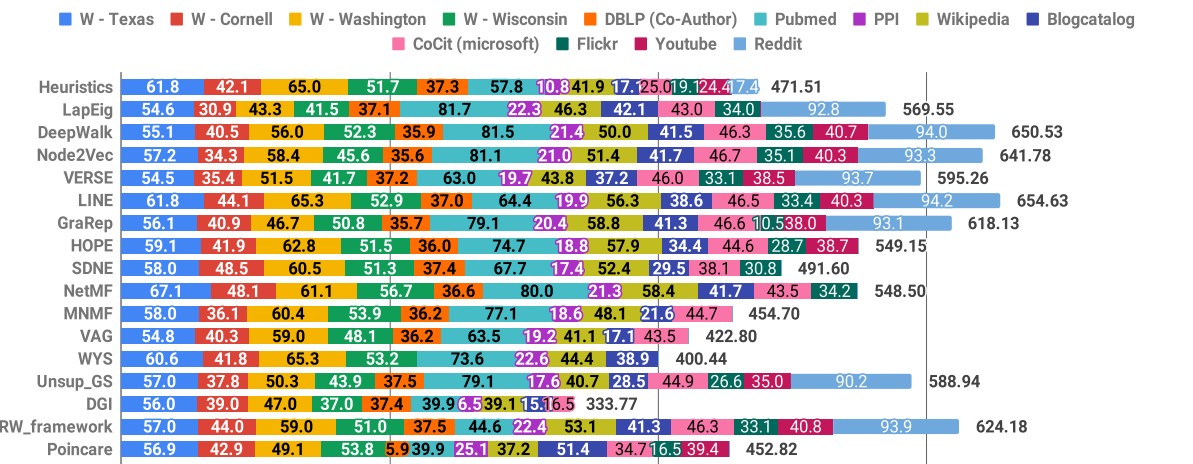

Figure 8: The node classification performance measured with Micro-f1 on train-test split of 50:50 with Logistic Regression. For each method, the number at the end of bar represent the summation of the Micro-f1 values across the datasets. The Micro-f1 score is multiplied by 100 for better readability.

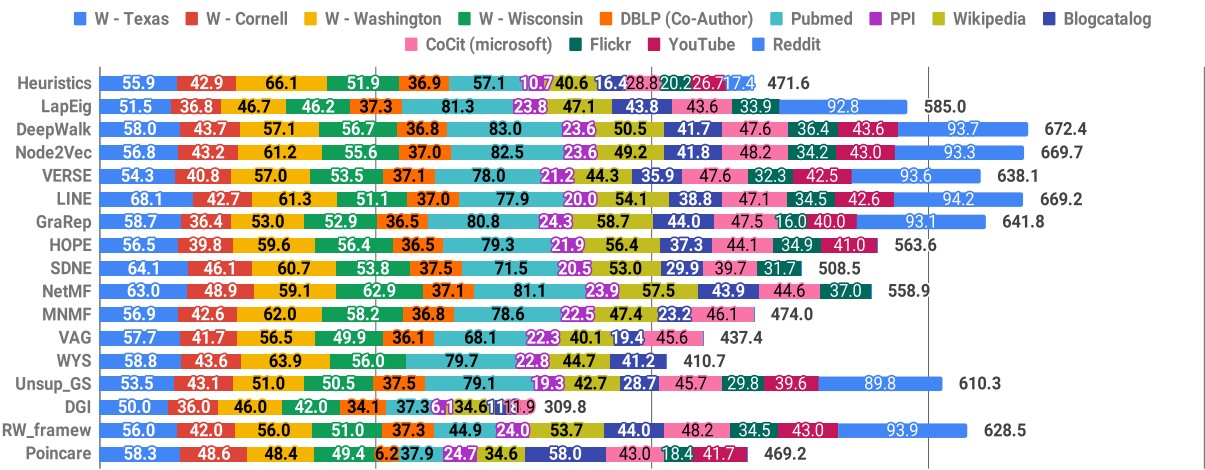

Figure 9: The node classification performance measured with Micro-f1 on train-test split of 50:50 with non-linear classifier. The Micro-f1 score is multiplied by 100 for better readability.

– larger values of standard error corresponds to a large variance in AUROC scores across 5 splits. In Figure 7, we report the distribution of average AUROC standard error of each embedding method over all datasets. We observe that most of the methods' median of box-plots is close to zero, suggesting that all methods are reasonably stable with MNMF and DeepWalk seemingly being the most robust across different splits and datasets.

**G. Dataset specific analysis:** There is a strong negative correlation between the sparsity of the dataset and the average link prediction performance across methods on said dataset (Pearson correlation (-0.82)). Clustering coefficient plays a secondary factor on aggregate link prediction performance – methods achieve poor link prediction performance on low clustering coefficient datasets such as p2p-Gnutella and Pubmed (an exception is MNMF which is explained by the fact that it explicitly accounts for community structure). On the other hand, on datasets with high CC (e.g. DBLP) all methods perform well (above 90%).

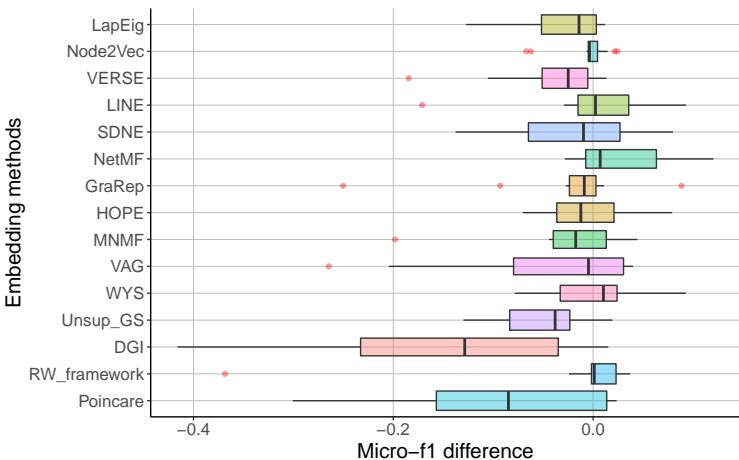

Figure 10: Illusion of Progress II: Node Classification: The Box-plot represents the distribution of differences between Micro-f1 of the embedding method and Deepwalk.

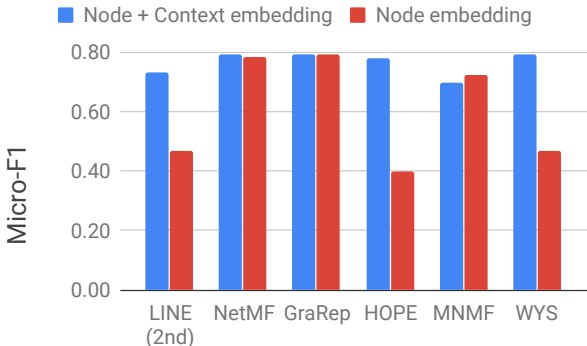

Figure 11: Node Classification on Directed Dataset (PubMed) with/without concatenation of Node embeddings and Context Embeddings (128 dimensions).

Next, we analyze the performance of embedding method HOPE on directed datasets. HOPE learns node embeddings by preserving asymmetric transitivity and hence is expected to perform well on link prediction tasks for the directed datasets. We observe that the HOPE method achieves competitive performance on two directed datasets (Wiki-vote and Epinions). However, HOPE achieves relatively low link prediction performance on other two directed datasets (Pubmed and P2P-Gnutella) with low clustering coefficient suggesting preserving asymmetric transitivity might not be "sufficient" for better link prediction performance on directed datasets with low clustering coefficient.

**H. Default vs Tuned Parameters:** In Figure 4, we show the impact of tuning the parameters of network embedding method on the link prediction performance. A positive difference implies the tuned parameters result in better performance than default parameters. Not surprisingly we see that for most methods tuning its parameters results in improved performance (the exceptions being NetMF and LINE for which default parameters work very well). The tuning of parameters for methods such as SDNE, MNMF, VAG, and WYS lead to a significant boost in the average AUROC score. This analysis underscores the importance of parametric tuning on downstream performance which is often ignored by authors when comparing with strawman methods and is not considered even in some benchmarking studies Khosla et al. (2019).

## 7.2 Node Classification

The node classification performance of 15 embedding methods measured in terms of Micro-f1 scores on 16 datasets with train-test split of 50:50 is reported in Figure 8 and Figure 9. We tune the hyper-parameters of each embedding method – mentioned in Section 2 – and report the best Micro-f1 score. We perform the node classification with both normalized and unnormalized embeddings and report the best performance. We make the following observations.

**A. Illusion of Progress II**: Here we compare and contrast the performance of recently proposed algorithms for network representation learning with Deepwalk on the node classification task in Figure 10. A positive difference here implies that the embedding method's performance is better than Deepwalk in Micro-f1. The improvement over Deepwalk is statistically significant (paired t-test) for none of the methods at significance level 0.05 when evaluated on non-attributed graphs. Somewhat startlingly, we observe that not a single method offers a consistent, statistically significant improvement over Deepwalk on this task. The closest one to being significant is NetMF, with a p-value of 0.07.

**Efficacy of Deepwalk on Node Classification task**: The efficacy of Deepwalk on node classification task can be attributed to i. tuning of Deepwalk's parameters and ii. tuning of logistic regression parameters. The default vs tuned parameters of Deepwalk method result in vast performance differences. For instance, with Logistic regression and train:test split of 50:50, we observe that the default vs tuned parameters result in following Micro-f1 scores: 0.19 vs 0.21 (PPI), 0.47 vs 0.50 (Wikipedia), 0.34 vs 0.36 (Flickr). This experiment highlights the importance of careful tuning of comparative methods.

**B. Analysis of the methods for node classification:** Although not better than Deepwalk (w.r.t statistical significance), NetMF achieves the highest overall performance for node classification with both linear and non-linear classifiers. LINE, DeepWalk, and Node2Vec are also competitive baselines for the task of node classification as their overall performance is closest to that of NetMF. The performance of GraRep on datasets with more labels is comparable with other methods (except on the Flickr dataset). The results for GraRep on the Flickr dataset are with an embedding dimensionality of 64 as increasing the dimensionality to 128 and 256 resulted in an Out-Of-Memory error on a modern machine with 500GB RAM and 28 cores. Note that the methods NetMF, MNMF, SDNE, VAG, and WYS did not finish execution on YouTube and Reddit. While scalability is currently outside our study's scope, the scalability of such methods is under active development (we refer the interested reader elsewhere Qiu et al. (2019); Liang et al. (2018a)).

**C. Laplacian Eigenmaps Performance:** We observe that Laplacian Eigenmaps achieves competitive Micro-f1 scores on several datasets. For instance, on the Blogcatalog dataset with 39 labels, Laplacian Eigenmaps achieves the best Micro-f1 score of 42.1% while on the Pubmed dataset, it outperforms all other embedding methods. With a non-linear classifier, Laplacian Eigenmaps achieves the second-best performance on the PPI dataset with 23.8% Micro-f1. The observed results for Laplacian Eigenmaps on evaluated datasets are better than those reported elsewhere Goyal & Ferrara (2018); Grover & Leskovec (2016) for both node classification and link prediction tasks. This improvement is due in part to the careful hyperparameter tuning of parameters of the corresponding classifier.

**D. Context embeddings can improve performance:** We find that levering both node and context embeddings of Skip-gram based models results in significant improvement (up to 25%) for most of the methods (see Figure 11). On the Pubmed dataset, we observe that the node classification performance of embedding methods, LINE ($2^{nd}$ order), HOPE, and WYS, is significantly lower than those of other methods. We found that the Pubmed dataset consists of around 80% sink nodes. As a result, when the Skip-gram based embedding methods generate node embeddings, the sink nodes appear most times as "context" nodes and rarely appear as "source" nodes. Hence, the node embeddings of sink nodes are of lower quality.

**E. Impact of non-linear classifier:** We study the impact of using a non-linear classifier (as opposed to a linear classifier) on node classification performance. The results are shown with a box plot in Figure 12. The box plot represents the distribution of differences of Micro-f1 scores computed with the non-linear (EigenPro Ma & Belkin (2017)) and linear classifiers (Logistic Regression). A positive difference implies that the non-linear classifier results are better than that of the linear classifier. For Verse, we see a 15% absolute

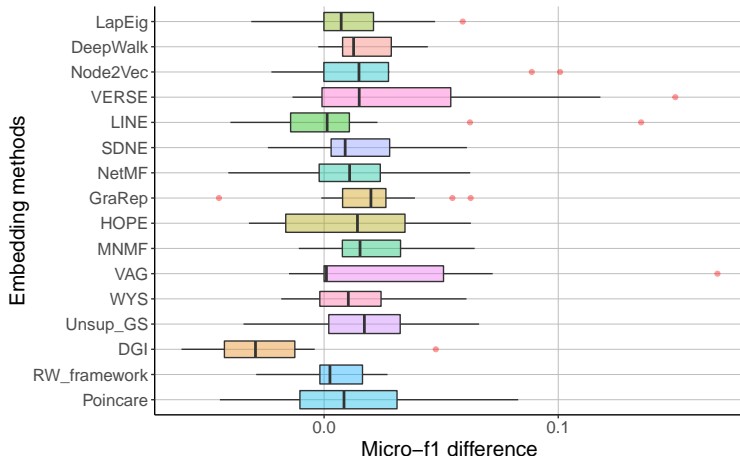

Figure 12: Box-plot represents the distribution of differences between Micro-f1 of the non-linear classifier and Micro-f1 of linear classifier on all the datasets.

increase when using a non-linear classifier on the PubMed dataset. The positive difference is statistically significant (with paired t-test) for methods DeepWalk, Verse, SDNE, GraRep, MNMF, and Unsupervised GraphSage with significance level 0.05. It is worth pointing out that this gain is less evident on the smaller datasets while on the larger datasets (more training data), the benefits of using a non-linear classifier are much clearer.

**F. Classifier Robustness:** We also note that the non-linear classifier appears to be more stable to the impact of normalization (of the embedding), as shown in Figure 13. A positive difference implies the normalization of embeddings improves performance.

**G. Dataset specific analysis** First, we observe that on datasets with fewer classes (up to five), the node classification heuristic is competitive - its performance is better than *several other embedding methods such as Laplacian Eigenmaps, Node2vec, Verse, GraRep, VAG, WYS and Unsupervised GraphSage.* However, as the number of classes in the datasets increases, we observe that the relative performance of the heuristic decreases drastically. The reduction in overall performance reflects that the node features that currently make up the baseline lack the discriminative power to classify multiple labels. Second, We observe that network embedding methods generally achieve poor node classification performance – reflected with low absolute values of Micro-f1 scores – on datasets with low homophily (see PPI and Flickr). On the contrary, on datasets with high homophily (Reddit and Pubmed), the network embedding methods achieve better node classification performance (with Micro-f1 scores above >80%). The Pearson correlation coefficient between average node classification performance on a dataset and its homogeneity is 0.86. Third, for multi-label datasets, clustering coefficient plays an important role along with homophily for overall high node classification performance (for example while both Wikipedia and PPI exhibit low homophily - Wikipedia has much higher clustering coefficient suggesting that local density plays a role in improved overall performance).

**H. Default vs Tuned Parameters:** In Figure 14 we again examine the impact of tuning parameters on node classification performance. Not surprisingly we once again observe that tuning the method's parameters results in a better Micro-f1 score for most methods (exceptions here are HOPE and GraRep). Tuning the parameters results in a huge gain in the Micro-f1 score of Laplacian Eigenmaps, VERSE, SDNE, NetMF, MNMF, VAG, WYS, and Unsupervised GraphSage. We reiterate the points we made in Section 7.1.H.

## 8 Discussion & Concluding Remarks

To conclude, among the methods we analyzed, we found that matrix factorization based approaches offer a small advantage over other methods on both link prediction and node classification. However, for the link prediction task, outside of MNMF and WYS, no other method offers a statistically significant improvement

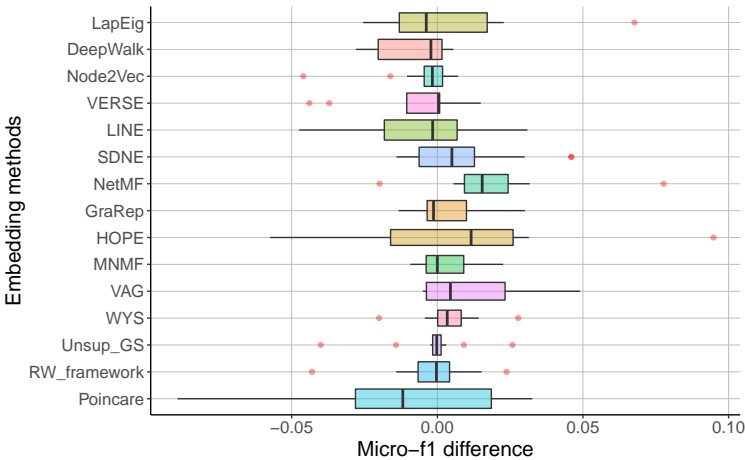

(a) Linear classifier : Logistic Regression

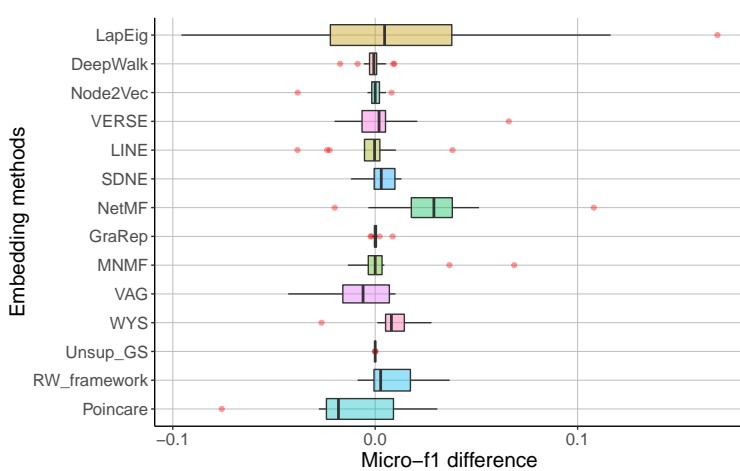

(b) Nonlinear classifier : EigenPro

Figure 13: Classifer Robustness: Box-plot represents the distribution of differences between Micro-f1 achieved with normalized embedding and Micro-f1 achieved with unnormalized embedding on all the datasets.

over DeepWalk on either task. Regarding task-specific baselines, the link prediction heuristic is simple, efficient to compute, and offers competitive performance, while the node-classification heuristic is effective only on datasets with fewer labels. Moving forward, this study suggests that the field of graph representation learning needs a renewed bearing and that careful benchmarking is essential for understanding true scientific advances. Said benchmarking should include more algorithms (existing and new ideas), non-trivial heuristics and methods for end-to-end tasks, as well as more evaluation datasets – ideally examined through a statistical lens, on a common platform (our study relies on authors' original code in various programming environments - in part because we wished to be true to the original implementations).

**Usability of our Study:** Network embedding methods has seen applications in multiple domains including drug-disease association prediction (Zhang et al., 2018b), drug–drug interaction prediction (Zhu et al., 2013), protein function prediction (Leiserson et al., 2017), for relation extraction in natural language processing(Jat et al., 2018), neural machine translation (Bastings et al., 2017), for outlier analysis (Zhao & Saligrama, 2009; Liang et al., 2018b), and entity resolution (Cohen & Richman, 2002; Getoor & Machanavajjhala, 2012). Our benchmarking study examines the performance of a range of methods on datasets drawn from many of these application domains. Moreover our study provides useful insights to researchers with respect to

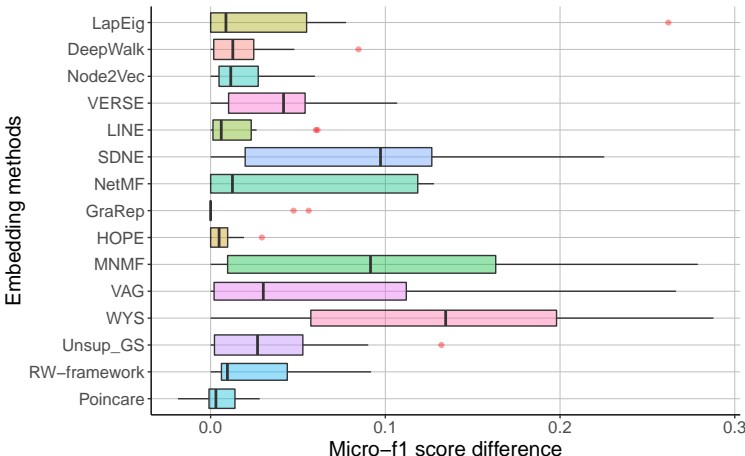

Figure 14: Default vs Tuned: Node Classification: The Box-plot represents the distribution of differences between Micro-f1 of the embedding method with default parameters vs tuned parameters.

which baseline methods and hyper-parameter settings to consider in these domains for downstream machine learning tasks such as link prediction and node classification.

**Future directions:** Scaling several of these methods is also a useful direction to pursue as this could open up benchmarking on larger more meaningful and practical datasets and end applications (Qiu et al., 2019; Liang et al., 2018a). Accounting for model complexity in this context becomes important. Note that Deepwalk is one of the simpler models (and easy to parallelize) and as we demonstrate can still perform in a competitive manner with respect to more complex models. Finally, we believe that the experimental data (375K runs) from this study can also serve as useful training data for improving auto-hyper-parameter tuning techniques (Tu et al., 2019a). In our initial analysis, we find that such techniques rarely match the best performance revealed by our comprehensive grid search for each method on each dataset.

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

# A   Appendix

## A.1   Additional experiments and analysis

In our study, in addition to Micro-F1 scores, we have computed Macro-F1 scores for all experimental settings (trends are similar to the reported results for Micro-F1). For node classification, we varied train: test splits from 10:90 to 90:10 (the trends are similar to what we have reported for the 50:50 split - in that no method outperforms DeepWalk with a significance level of 0.05). We also studied the impact of embedding normalization on the link prediction task (the impact of normalization was negligible with the logistic classifier, unlike what we observed for the node classification task). Additionally, we analyzed the impact of embedding dimension on both the tasks. We also study the impact of binary function on the link prediction task. These additional results and analysis will be available in a technical report.

# B   Summary of the methods

In this section, we give a summary of the network embedding methods evaluated in our work. Herein, for each models along with their description.

**Laplacian Eigenmaps Belkin & Niyogi (2003)**: Laplacian Eigenmaps generates a $d$-dimensional embedding of the graph using the smallest $d$ eigenvectors of Laplacian matrix $L = D - A$.

$$\underset{U}{\text{minimize}} \quad trace(U^T LU)$$
$$\text{subject to} \quad U^T DU = I$$

$U$ is generated embedding matrix $\mathbb{R}^{|V|*d}$. The above equation can be reduced to simple minimization of L2 distance for adjacent nodes - $\Sigma_{i,j}||u^{(i)} - u^{(j)}||^2 A_{ij}$. Laplacian Eigenmaps levers the first order information for generating the embeddings.

**DeepWalk Perozzi et al. (2014)**: DeepWalk is a random walk based network embedding method which uses truncated random walks and levers local information from these generated walks to learn similar latent representations. DeepWalk draws inspiration from Skip-gram model in Word2vec Mikolov et al. (2013) by treating random walks as sequences and optimizing following objective function:

$$minimize_\Phi \ -\log Pr(\{v_{i-w}, \cdots, v_{i+w}\}\backslash v_i \mid \Phi(v_i)) \tag{1}$$

where $v_i$ is target node while $\{v_{i-w}, \cdots, v_{i+w}\}\backslash v_i$ are the context nodes. $\Phi(v_i)$ denotes the embedding of the node $v_i$. Since the objective function is expensive to compute for large graphs, it is approximated using Hierarchical SoftmaxMnih & Hinton (2009).

**Node2Vec Grover & Leskovec (2016)**: Node2Vec is a biased random walk based network embedding method which allows the random walk to be more flexible in exploring the graph neighborhoods. The flexibility of the random walk is achieved by interpolating between Breadth-first traversal and Depth-first traversal. The objective function is again based on Skip-gram model -based on Word2vec Mikolov et al.

(2013), and since the objective function is expensive to compute for large graphs, it is approximated by negative sampling Mikolov et al. (2013).

**GraRep Cao et al. (2015)**: GraRep is a matrix factorization based network embedding method which captures the global structural information of the graph while learning node embeddings. The authors observe that the existing Skip-gram based models project all the $k$-step relational information into a common subspace and then, argue the importance of preserving different $k$-step relational information in separate subspaces. The loss function to preserve the $k$-step relationship between node $w$ and $c$ is proposed as:

$$
\begin{aligned}
L_k(u,v) =& A_{u,v}^k \cdot \log \sigma(\Phi(u) \cdot \Phi(v)) \\
&+ \frac{\lambda}{|V|} \sum_{v' \in V, v' \neq v} A_{u,v'}^k \cdot \log \sigma(-\Phi(u) \cdot \Phi(v'))
\end{aligned}
\tag{2}
$$

where $v'$ refers to the negative node at $k$-th step for node $u$ (see Table 1 for additional notation, e.g. $\lambda$). The above loss function in closed form results in log-transformed, probabilistic adjacency matrix which is factorized with SVD for generating each $k$-step representation. The final node representation is generated by concatenation of all the $k$-step representations.

**NetMF Qiu et al. (2018)**: NetMF is a matrix factorization based network embedding method. NetMF presents theoretical proofs for their claim that Skip-gram models with negative sampling are implicitly approximating and factorizing appropriate matrices constructed with the help of graph Laplacians. The objective matrix based for NetMF on small context window T is given by (see Table 1 for notation):

$$
\log \left( \text{vol}(G) \left( \frac{1}{T} \sum_{r=1}^{T} \left( D^{-1}A \right)^r \right) D^{-1} \right) - \log b
\tag{3}
$$

where $\text{vol}(G)$ refers to sum of all edge weights and $b$ corresponds to number of negative samples in skip-gram model. NetMF factorizes the above closed form DeepWalk matrix with SVD in order to generate node embedding and provides two algorithms for small context window and large context window.

**M-NMF Wang et al. (2017)**: M-NMF is a matrix factorization based network embedding method which generates node embeddings that preserves the microscopic information in form of first-order and second-order proximities among nodes and the generated embeddings also preserve mesoscopic information in form of community structure. The objective function for M-NMF is given as

$$
\begin{aligned}
O = \min_{U,V C,H \geq 0} & \|S - VU^T\|^2 + \alpha * \|H - UC^T\|_F^2 \\
& - \beta tr(H^T B H) + \zeta \|HH^T - I\|_F^2
\end{aligned}
\tag{4}
$$

where $H$ is the binary community membership matrix, $C$ is the latent representations of communities and $B$ is the modularity matrix obtained from the adjacency matrix, $A$ (see Table 1 for rest of the notations). Overall, M-NMF discovers communities through modularity constraints. The node embeddings generated with the help of microscopic information and community embeddings are then, jointly optimized by assuming consensus relationship between both node and community embeddings.

**HOPE Ou et al. (2016)**: HOPE is a matrix factorization based network embedding method which generates node embeddings that preserve asymmetric transitivity of nodes in directed graphs. If there exists a directed edge from node $u$ to $w$ and $w$ to $v$, then – due to asymmetric transitivity property – an edge from $u$ to $v$ is more likely to form than edge from $v$ to $u$. The objective function of HOPE is given as follows

$$
min \; \|S - U^s (U^t)^T\|_2^2
\tag{5}
$$

where $U^s$ and $U^t$ are the source and target embeddings. In order to preserve asymmetric transitivity of nodes, the proximity matrix $S$ is constructed using a similarity metric which respects the directionality of edges. The node embeddings are generated by factorizing the proximity matrix with generalized SVD Paige & Saunders (1981).

**LINE Tang et al. (2015)**: LINE is an optimization-based network embedding method which optimizes an objective function that preserves both first and second order proximity among nodes in the embedding space. The objective function for first-order proximity is given as:

$$O_1 = - \sum_{(u,v) \in E} A_{u,v} \; log \; \sigma(\Phi(u).\Phi(v)) \tag{6}$$

The objective function to preserve the second order proximity is given as:

$$O_2 = - \sum_{(u,v) \in E} A_{u,v} \; log \; \frac{exp(\Phi(u).\psi(v))}{\sum_{v' \in V, v' \neq v} exp(\Phi(u).\psi(v'))} \tag{7}$$

where $\psi(u)$ represents context embedding of node $u$ (see Table 1 for rest of the notations). The first-order proximity corresponds to local proximity between nodes based on the presence of edges in the graph while the second-order proximity corresponds to global proximity between nodes based on shared neighborhoods of those nodes in the graph. Since the objective function is expensive to compute for large graphs, it is approximated by negative sampling Mikolov et al. (2013).

**Verse Tsitsulin et al. (2018)**: Verse is an optimization-based network embedding method which optimizes an objective function that minimizes the Kullback-Leibler (KL) divergence from the given similarity distribution in graph space to similarity distribution in embedding space (E). The objective function is given as follows:

$$\sum_{v \in V} \text{KL}(sim_G(v, .) \| sim_E(v, .)) \tag{8}$$

The similarity distribution in graph space could be constructed with help of Personalized PageRankPage et al. (1999), SimRankJeh & Widom (2002), or Adjacency matrixTsitsulin et al. (2018). Since the objective function is expensive to compute for large graphs, it is approximated by Noise Constrastive Estimation Gutmann & Hyvärinen (2010).

**SDNE Wang et al. (2016a)**: SDNE is a deep autoencoder based network embedding method which optimizes an objective function that preserves both first and second order proximity among nodes in the embedding space. The objective function of SDNE is given below

$$\mathcal{L}_{joint} = \alpha \mathcal{L}_{1st} + \mathcal{L}_{2nd} + \nu \mathcal{L}_{reg} \tag{9}$$

where $\mathcal{L}_{1st}$ and $\mathcal{L}_{2nd}$ are loss functions to preserve first-order and second-order proximities respectively, while $\mathcal{L}_{reg}$ is the regularizer term. The authors propose a semi-supervised deep model to minimize the mentioned objective function. The deep model consists of two components: supervised and unsupervised. The supervised component attempts to preserve the first-order proximity while the unsupervised component attempts to preserve the second-order proximity by minimizing reconstruction loss of nodes.

**VAG Kipf & Welling (2016)**: VAG is a graph autoencoder based network embedding method which minimizes the reconstruction loss of the adjacency matrix. The reconstruction matrix is generated as $\hat{A} = \sigma(ZZ^T)$ where $Z$ is node embeddings generated with Graph Convolutional Networks (GCN) Kipf & Welling (2017) as $Z = GCN(X, A)$ with $X$ as node features (see Table 1 for additional notation). In the case of unattributed graphs, the node feature matrix is the identity matrix.

**Watch Your Step Abu-El-Haija et al. (2018)**: Watch Your Step (WYS) addresses the sensitivity issue of hyper-parameters in the random walk based embedding methods. WYS solves the sensitivity issue with the attention mechanism on the expected random walk matrix. The attention mechanism guides the random walk to focus on short or long term dependencies pertinent to the input graph. The objective function of WYS is given as

$$\min_{L,R,q} \quad \beta \|q\|_2^2 - \|\mathbb{E}[D; q] \circ \log(\sigma(L * R^T))$$
$$- \mathbb{1}[A = 0] \circ \log(1 - \sigma(L * R^T))\|_1 \tag{10}$$

where $q$ is attention parameter vector, $L$ and $R$ are node embeddings, $E[D; q]$ is expectation on the random walk matrix (see Table 1 for rest of the notations).

**Unsupervised-GraphSage Hamilton et al. (2017a)**: Graphsage is a message-passing based graph neural method that learns to aggregate feature information from a node's local neighborhood. The neighborhood of the node $u$ is defined through a parameter $k$ and the nodes that are $k$ hops from $u$ are considered as its neighbors. Due to this definition of node neighborhood, one can train GraphSage in a minibatch setting. The objective function of Unsupervised-GraphSage is shown below

$$- log(\sigma(z_u^T, z_v)) - Q.\mathbb{E}_{k \sim P_n(v)} log(\sigma(-z_u^T, z_k)) \tag{11}$$

where $z_u$ is the node representation of node u, $Q$ is the number of negative samples and $P_n(v)$ is negative sampling distribution.

**DGL Velickovic et al. (2019)**: Deep Graph Infomax (DGI) Velickovic et al. (2019) is a recent and highly popular unsupervised node representation learning approach that has achieved excellent performance for graphs with node attributes. DGI uses an InfoMax based objective that aims to maximize the mutual information between a node's input features based patch-level representation and its corresponding graph-level representation. This enforces the message passing based patch encoder to prefer similarity information shared across patches. The objective function of DGI is given as

$$\frac{1}{N + M} \Big( \sum_{i=1}^{N} \mathbb{E}_{(X,A)} \left[ log \, D(h_i, s) \right] + \sum_{j=1}^{M} \mathbb{E}_{(\tilde{X}, \tilde{A})} \left[ log \, (1 - D(\tilde{h_j}, s)] \right) \Big) \tag{12}$$

where $N$ and $M$ are the number of nodes and number of negative samples respectively. $X, A$ and $\tilde{X}, \tilde{A}$ are features and adjaceny matrix of original and alternate graph, respectively. $s$ is summary vector of the graph. $h_i$ and $\tilde{h_i}$ is the hidden representation of the node $i$ in original and alternate graph, respectively.

**RandomWalk-Framework Huang et al. (2021)**: Huang et al. proposed an analytical framework for categorizing and proposing novel random-walk based node embedding methods. The framework consists of three components i)random-walk process, ii) similarity function, and iii) embedding algorithm. The authors utilize standard random-walk process for undirected graphs and Pagerank process for directed graphs. For similarity function, the framework relies on Pointwise Mutual Information and autocovariance metrics. The embedding methods in the framework are matrix factorization and sampling.

**Poincaré Nickel & Kiela (2017)**: Nickel et al. Nickel & Kiela (2017) propose that the embeddings learned in the hyperbolic space can better reflect the semantic similarity of symbolic data than the embeddings learned in euclidean space, especially when the data has hierarchical structure. The loss function of Poincaré is described as

$$\mathcal{L}(\theta) = \sum_{(u,v) \in E} log \frac{e^{-d(u,v)}}{\sum_{v' \in N_v'} e^{-d(u,v')}} \tag{13}$$

where $d(u,v) = arccosh \left( 1 + 2\frac{||u-v||^2}{(1-||u||^2)(1-||v||^2)} \right)$. The equation 13 is optimized via stochastic Riemannian optimization methods.

## C   Link Prediction Dataset Statistics

The statistics of the five folds of link prediction dataset is shown in Table 4 and Table 5. The link prediction folds are formed by following the procedure specified in section 6.1. The train graph has around 80% edges while the test graph has 20% edges. If the test edge contains node (say $t$) that is not present in training graph, we delete that test edge as we cannot learn the node embedding of node $t$ during training. The flowchart for graph construction is shared in Figure 15a while the flowchart for creation of train and test sets for Link prediction classifier is shared in Figure 15b.

| Dataset | Fold | Num Nodes | Num Edges | Train Graph Nodes | Train Graph Edges | Test Graph Nodes | Test Graph Edges |
|---|---|---|---|---|---|---|---|
| texas | 0 | 186 | 464 | 159 | 224 | 39 | 28 |
| texas | 1 | 186 | 464 | 162 | 216 | 48 | 35 |
| texas | 2 | 186 | 464 | 168 | 217 | 53 | 42 |
| texas | 3 | 186 | 464 | 157 | 216 | 40 | 30 |
| texas | 4 | 186 | 464 | 156 | 211 | 44 | 32 |
| cornell | 0 | 195 | 478 | 163 | 222 | 45 | 34 |
| cornell | 1 | 195 | 478 | 161 | 216 | 53 | 37 |
| cornell | 2 | 195 | 478 | 160 | 217 | 43 | 34 |
| cornell | 3 | 195 | 478 | 155 | 215 | 44 | 29 |
| cornell | 4 | 195 | 478 | 160 | 213 | 51 | 40 |
| washington | 0 | 230 | 596 | 203 | 291 | 79 | 61 |
| washington | 1 | 230 | 596 | 191 | 285 | 74 | 52 |
| washington | 2 | 230 | 596 | 187 | 276 | 71 | 60 |
| washington | 3 | 230 | 596 | 180 | 272 | 63 | 46 |
| washington | 4 | 230 | 596 | 196 | 287 | 69 | 56 |
| wisconsin | 0 | 265 | 724 | 232 | 365 | 85 | 65 |
| wisconsin | 1 | 265 | 724 | 222 | 344 | 89 | 70 |
| wisconsin | 2 | 265 | 724 | 220 | 355 | 76 | 57 |
| wisconsin | 3 | 265 | 724 | 225 | 348 | 98 | 73 |
| wisconsin | 4 | 265 | 724 | 239 | 375 | 82 | 61 |
| ppi | 0 | 3,890 | 38,739 | 3771 | 30260 | 2771 | 7492 |
| ppi | 1 | 3,890 | 38,739 | 3744 | 30367 | 2733 | 7351 |
| ppi | 2 | 3,890 | 38,739 | 3768 | 30367 | 2738 | 7364 |
| ppi | 3 | 3,890 | 38,739 | 3753 | 30179 | 2755 | 7547 |
| ppi | 4 | 3,890 | 38,739 | 3758 | 30170 | 2765 | 7556 |
| wikipedia | 0 | 4,777 | 92,517 | 4777 | 73959 | 4563 | 18336 |
| wikipedia | 1 | 4777 | 92,517 | 4777 | 73702 | 4568 | 18593 |
| wikipedia | 2 | 4777 | 92,517 | 4777 | 73846 | 4558 | 18449 |
| wikipedia | 3 | 4777 | 92,517 | 4777 | 73666 | 4554 | 18629 |
| wikipedia | 4 | 4777 | 92,517 | 4777 | 73748 | 4552 | 18547 |
| wikivote | 0 | 7,115 | 103,689 | 6611 | 81239 | 3840 | 19955 |
| wikivote | 1 | 7,115 | 103,689 | 6596 | 81151 | 3845 | 20041 |
| wikivote | 2 | 7,115 | 103,689 | 6572 | 81154 | 3854 | 20042 |
| wikivote | 3 | 7,115 | 103,689 | 6558 | 81113 | 3854 | 20006 |
| wikivote | 4 | 7,115 | 103,689 | 6573 | 80993 | 3894 | 20100 |
| blogcatalog | 0 | 10,312 | 333,983 | 10242 | 267094 | 8958 | 66795 |
| blogcatalog | 1 | 10,312 | 333,983 | 10247 | 266883 | 8973 | 67013 |
| blogcatalog | 2 | 10,312 | 333,983 | 10227 | 267134 | 8952 | 66746 |
| blogcatalog | 3 | 10,312 | 333,983 | 10237 | 267025 | 8936 | 66862 |
| blogcatalog | 4 | 10,312 | 333,983 | 10252 | 266687 | 8955 | 67215 |
| co-author | 0 | 18,721 | 122,245 | 17691 | 82506 | 12349 | 19996 |
| co-author | 1 | 18,721 | 122,245 | 17669 | 82076 | 12550 | 20386 |
| co-author | 2 | 18,721 | 122,245 | 17697 | 82298 | 12513 | 20172 |
| co-author | 3 | 18,721 | 122,245 | 17695 | 82183 | 12559 | 20309 |
| co-author | 4 | 18,721 | 122,245 | 17704 | 82205 | 12482 | 20319 |
| pubmed | 0 | 19,717 | 44,338 | 17551 | 35347 | 6086 | 6643 |
| pubmed | 1 | 19,717 | 44,338 | 17541 | 35250 | 6100 | 6698 |
| pubmed | 2 | 19,717 | 44,338 | 17592 | 35321 | 6107 | 6692 |
| pubmed | 3 | 19,717 | 44,338 | 17531 | 35384 | 6028 | 6549 |
| pubmed | 4 | 19,717 | 44,338 | 17561 | 35417 | 6121 | 6585 |
| microsoft | 0 | 44,034 | 195,361 | 40423 | 155124 | 25118 | 36949 |
| microsoft | 1 | 44,034 | 195,361 | 40464 | 155046 | 25228 | 37048 |
| microsoft | 2 | 44,034 | 195,361 | 40344 | 155070 | 25172 | 36872 |
| microsoft | 3 | 44,034 | 195,361 | 40315 | 155133 | 25078 | 36806 |
| microsoft | 4 | 44,034 | 195,361 | 40365 | 155146 | 25123 | 36875 |
| p2p-gnutella31 | 0 | 62,586 | 147,892 | 56322 | 118298 | 22077 | 22869 |
| p2p-gnutella31 | 1 | 62,586 | 147,892 | 56235 | 118089 | 22092 | 22985 |
| p2p-gnutella31 | 2 | 62,586 | 147,892 | 56506 | 118524 | 21957 | 22828 |
| p2p-gnutella31 | 3 | 62,586 | 147,892 | 56295 | 118244 | 22079 | 22911 |
| p2p-gnutella31 | 4 | 62,586 | 147,892 | 56326 | 118207 | 22067 | 22980 |
| flickr | 0 | 80,513 | 5,899,882 | 80379 | 4718007 | 74083 | 1181588 |
| flickr | 1 | 80,513 | 5,899,882 | 80382 | 4720230 | 73991 | 1179368 |
| flickr | 2 | 80,513 | 5,899,882 | 80372 | 4720613 | 73966 | 1178954 |
| flickr | 3 | 80,513 | 5,899,882 | 80358 | 4718527 | 73984 | 1181007 |
| flickr | 4 | 80,513 | 5,899,882 | 80358 | 4718527 | 73984 | 1181007 |

Table 4: Link Prediction Folds Statistics

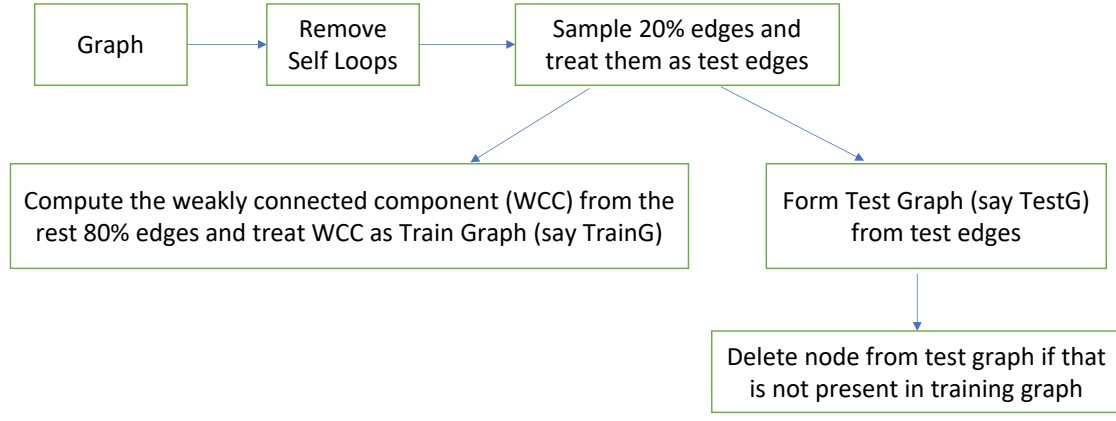

(a) Visual illustration for the graph construction.

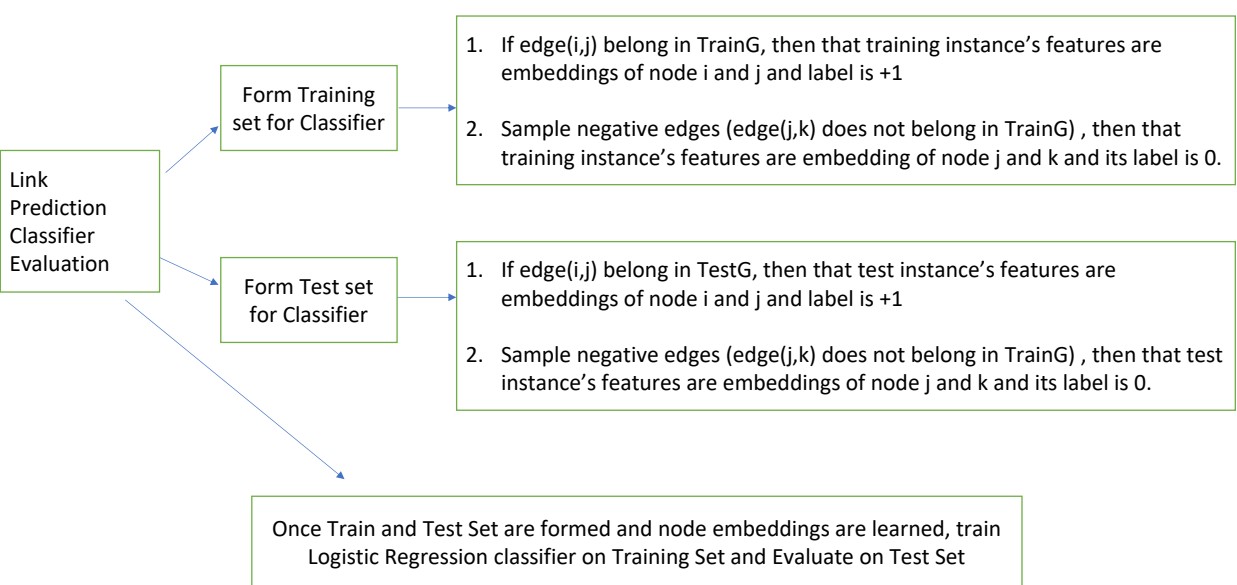

(b) Visual illustration for training and test set creation for Link Prediction classifier.

## D  Link Prediction: Ranking

We rank the methods based on the average AUROC score and compare the ranking of the methods based on evaluation strategy – dot vs. classifier. The results are presented in Table 6. If the average AUROC between the two methods is within 0.5 ($\times$ 100) points, then they share the same ranks. We observe that in case of the random-walk based methods such Deepwalk, Node2vec, RW_framework as well as the family of methods covered by NetMF Qiu et al. (2018) (LINE, Deepwalk, Node2vec), the use of classifier improves the link prediction performance. Moreover, even though the rank for VAG increase, the overall average AUROC of VAG still improves. For the rest of the methods, the ranking is similar (rank difference $\leq 4$) with respect to the evaluation strategy.

## E  Link Prediction: Mean Average Precision

The link prediction performance of 13 embedding methods measured in terms of mean average precision on 15 datasets is shown in Figure 16. We observe that the claim of illusion of progress for the link prediction task holds with mean average precision score.

| Dataset | Fold | Num Nodes | Num Edges | Train Graph Nodes | Train Graph Edges | Test Graph Nodes | Test Graph Edges |
|---|---|---|---|---|---|---|---|
| epinions | 0 | 75,879 | 508,837 | 67900 | 340192 | 28079 | 90340 |
| epinions | 1 | 75,879 | 508,837 | 68049 | 340091 | 28036 | 90300 |
| epinions | 2 | 75,879 | 508,837 | 67887 | 340079 | 27923 | 90277 |
| epinions | 3 | 75,879 | 508,837 | 68104 | 340273 | 27949 | 90156 |
| epinions | 4 | 75,879 | 508,837 | 67928 | 340080 | 27913 | 90288 |
| youtube | 0 | 1,134,890 | 2,987,624 | 966732 | 2362025 | 260866 | 446980 |
| youtube | 1 | 1,134,890 | 2,987,624 | 966749 | 2363358 | 260352 | 445760 |
| youtube | 2 | 1,134,890 | 2,987,624 | 967081 | 2362951 | 260846 | 446739 |
| youtube | 3 | 1,134,890 | 2,987,624 | 966025 | 2362875 | 260048 | 445527 |
| youtube | 4 | 1,134,890 | 2,987,624 | 967130 | 2364336 | 260061 | 445068 |
| reddit | 0 | 231,443 | 11,606,919 | 230562 | 9282990 | 221639 | 2322903 |
| reddit | 1 | 231,443 | 11,606,919 | 230607 | 9285995 | 221643 | 2319955 |
| reddit | 2 | 231,443 | 11,606,919 | 230576 | 9282918 | 221573 | 2322994 |
| reddit | 3 | 231,443 | 11,606,919 | 230583 | 9286122 | 221524 | 2319805 |
| reddit | 4 | 231,443 | 11,606,919 | 230593 | 9284390 | 221609 | 2321546 |

Table 5: Link Prediction Folds Statistics

| | Avg. AUROC: Dot Product | Avg. AUROC: Classifier | Dot-product Ranking | Classifier Ranking |
|---|---|---|---|---|
| LaplacianEmb | 70.65 | 83.33 | 14 | 13 |
| Deepwalk | 82.10 | 89.89 | 6 | 3 |
| Node2vec | 84.23 | 89.77 | 4 | 3 |
| Grarep | 77.89 | 88.44 | 11 | 9 |
| NetMF | 76.08 | 89.23 | 12 | 3 |
| MNMF | 88.71 | 95.25 | 2 | 1 |
| LINE | 80.36 | 89.33 | 9 | 3 |
| Verse | 88.01 | 89.33 | 3 | 3 |
| VAG | 81.51 | 86.53 | 7 | 12 |
| WYS | 89.98 | 90.45 | 1 | 2 |
| SDNE | 74.17 | 81.99 | 13 | 14 |
| HOPE | 83.71 | 88.21 | 5 | 9 |
| Unsup_GS | 81.50 | 87.22 | 7 | 11 |
| RW_framework | 80.40 | 89.35 | 9 | 3 |
| Poincaré | 59.93 | 76.61 | 15 | 15 |

Table 6: Ranking of the method based on the average AUROC score. AUROC score is is multiplied by 100 for better readability. If the average AUROC between the two methods is within 0.5 ($\times$ 100) points, then they share the same ranks.

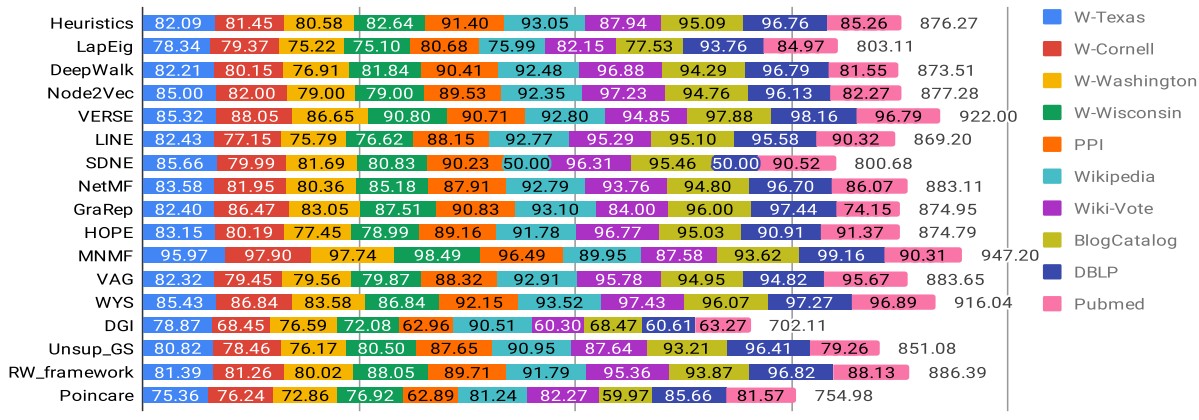

(a) Smaller datasets: All methods complete execution.

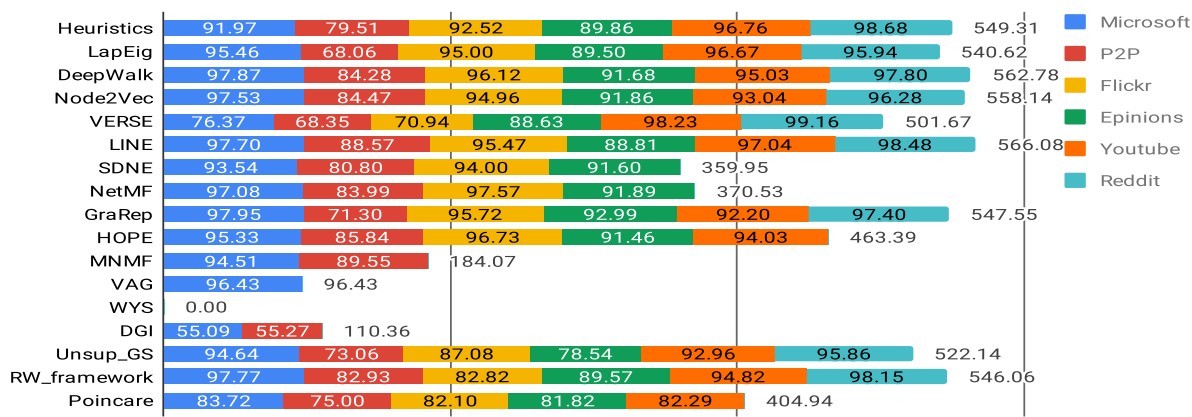

(b) Larger datasets: Not all methods complete execution.

Figure 16: Link Prediction performance measured with **Mean Average Precision (MAP)**. The MAP score is multiplied by 100 for better readability. Not all methods are scalable on the datasets shown in Figure 16b.

## F   Default Parameters

Figure 7 shows the parameters considered to be default in our experiments in Section 4 and Section 14.

| Methods | Default Parameters |
| --- | --- |
| LapEig | d=128 |
| Deepwalk | d=128, walk length=40, number of walks=80, window size=10 |
| Node2vec | d=128, walk length=40, number of walks=80, window size=10, p=4.0, q=1.0 |
| GraRep | d=128, k=4 |
| NetMF | d=128, T=10, $\lambda$=3, H=128 |
| MNMF | d=128, alpha=0.1, beta=10 |
| Line | d=128, samples=10 billion |
| Verse | d=128, alpha=0.85, negs=3 |
| VAG | d=128, h=128 |
| WYS | d=128, hops=5 |
| SDNE | d=128, alpha=0.2, beta=5 |
| Hope | d=128, alpha= spectral radius |
| Unsup_GS | d=128, walk length=40, number of walks=80, |
| RW_framework | d=128, algorithm=factorization, markov-time=100, similarity=PMI |
| Poincaré | d=128, manifold=poincare, negative samples=50,model=distance |

Table 7: The parameters considered to be default on our experiments in Section 4 and Section 14.

