# OpenReview forum: "Benchmarking and Analyzing Unsupervised Network Representation Learning and the Illusion of Progress"
_TMLR — Accepted by TMLR_

### Review · Reviewer_tvky · 2022-04-25

**Summary Of Contributions:**

This paper provides extensive benchmark studies of various unsupervised graph representation learning methods across many different datasets. The main focus of this study is to identify whether the advanced methods can outperform the simple DeepWalk baseline, and the authors conclude that most of the recent methods cannot do so, especially on the link prediction task. Also, the suggested simple network properties-based methods show decent performances on node classification and link prediction tasks, thus the authors suggest the community use such simpler models as baselines.

**Broader Impact Concerns:**

While there are no explicit sections for a broader impact statement in this paper, I think there are no concerns about the ethical implications of this work.

**Requested Changes:**

# Suggestions
**that critically affect my recommendation**
* The authors should evaluate the embedding space of compared methods. Does the embedding space reflect the heuristic properties of nodes and edges, for example, node degrees or centralities? Performances alone are not enough to benchmark several models. Also, it is recommended to analyze the embedding space with spectrum analysis to identify the important signals.
* The authors should re-evaluate different methods: tuning the hyperparameters with validation sets and then reporting the final performances with test sets, not equally using test sets for both.
* The authors should include the results on more recent OGB datasets [1] and more recent unsupervised learning methods [2, 3, 4].
* The authors should include the discussions on efficiencies of different models, to see the performance gains with models' complexities.

---

# Suggestions
**that would strengthen the work in my point of view**
* It is better to change the title. The main focus of this work is the benchmark study of different unsupervised network representation methods, and I think there should be the word benchmark in the title. When I first saw the title, for me, it is hard to understand this paper is relevant to the benchmark study.
* The description of every method in Table 2 is not enough. This paper is not self-contained, as the readers who are not familiar with the compared methods in Table 2 must jump to the original paper to understand the methods.
* The authors provide only the ranges of the candidate hyperparameters. It is better to provide also the searched hyperparameters that perform the best.
* It is better to visually illustrate the dataset preprocessing procedures in the "Construction of the train and test sets" paragraph of Section 6.1 Link prediction.

---

# Questions
**that should be also additionally clarified in the paper**
* Does the authors use the hyperparameters from the original paper, or arbitrarily search for the hyperparameters without including the original ones?
* What is the meaning of default parameters in H. Default vs Tuned Parameters?

**Strengths And Weaknesses:**

# Strengths
* The performed experiments for benchmarking unsupervised graph representation methods are extensive.
* The authors faithfully tune the hyperparameters for every model, to accurately evaluate different methods.

---

# Weaknesses
Overall, there should be more empirical results and analyses to benchmark different methods.
* The performance for two downstream tasks (i.e., node and link prediction) is not enough to evaluate various methods. The authors should also provide more in-depth analyses about the representation space. For example, does the learned representation space accurately reflect the important node property, such as degree, page rank, clustering coefficient, to name a few?
* The authors use the test dataset to tune the hyperparameters. Hyperparameter tuning should be done with validation sets.
* A lack of analyses on efficiencies. In particular, the performance gains from the method should be evaluated along with its time and memory complexity. Does the complex the model, the higher the performance?
* A lack of recent large-scale benchmark datasets, namely Open Graph Benchmark [1].
* There should be more recent unsupervised graph embedding methods to benchmark [2, 3, 4]. All the compared models outlined in Table 2 are the works before 2019.

---

[1] Hu et al. Open Graph Benchmark: Datasets for Machine Learning on Graphs. NeurIPS 2020.

[2] Hassani et al. Contrastive Multi-View Representation Learning on Graphs. ICML 2020.

[3] Peng et al. Graph Representation Learning via Graphical Mutual Information Maximization. WWW 2020.

[4] Mo et al. SimpleUnsupervisedGraphRepresentationLearning. AAAI 2022.

---

> ### Author Response · Authors · 2022-05-15
> **Response from the authors**
>
> We thank the reviewer for the comments and valuable feedback. Please find below our detailed response.
>
> We want to emphasize that the work of Khosla et al. 2019 and Goyal et al. 2018 is benchmarking. Our work - similar to the work of Hand 2006 - does benchmark results of a host of methods on a range of datasets but goes a step further than simple benchmarking.  It offers strong evidence that the community may need to rethink whether inventing yet another method in this space is actually yielding a significant improvement to our state of knowledge. Our results strongly suggest that improvements are largely an illusion, especially with respect to DeepWalk.
>
> Analyzing node properties and downstream tasks: We focus on node classification and link prediction task as it is frequently used by the community for the evaluation. We use the node properties of degree, pagerank, clustering coefficient, and other properties in our proposed heuristics and evaluate their performance on the downstream tasks. We are not sure how the vector representations could be used to reflect the node property of Pagerank.
>
> Tuning of the hyperparameters: We do not use the test dataset to tune the hyperparameters of the network models.
>  In the case of the node classification task, across all network datasets, we learn node representations of each dataset under different hyper-parameters in an unsupervised manner. Then for each hyper-parameter of a model, we evaluate the learned node embeddings using a classifier and report results averaged over 10 different train/test splits. For tuning any available hyper-parameters of the considered classifier, we further use 5-fold cross-validation in each of these ten train/test data splits. Finally, we report the best-averaged test results among all model hyperparameters.
> In the case of the link prediction task, across all network datasets, for 5 different train/test network data split, we learn node representations of each training data under different hyper-parameters in an unsupervised manner. Then for each train/test split of a dataset, we evaluate the learned node embeddings to predict missing edges present in the test set. For this, we either use a dot product or a classifier. In the case of using a classifier, its hyperparameters are tuned using a 5-fold cross-validation set. Finally, we report the best-averaged test results across all hyper-parameters for each model.
>
> OGB dataset: We have selected methods and datasets that are widely used (highly cited) for the task of unsupervised node representation learning, especially for the featureless datasets. Note that our selected datasets share few characteristics with the OGB dataset. For instance, the co-author/dblp network has the same domain as that of arxiv and papers100M, while PPI has the same domain as that of ogbn-proteins. Moreover, OGB datasets are huge graphs and most of the methods cannot operate on large graphs – as evident from experiments on Flickr and Reddit datasets.
>
> Comment on the Methods: We have included a recent unsupervised learning method [1]. The suggested references 2, 3, and 4 require node features while our selected graphs do not contain features.
>
> Model complexity: We would like to reiterate that model complexity and scalability is not the focus of our discussion. Our focus in this work is on if we are making progress in the graph representation learning and results suggest that we aren’t since Deepwalk was published in KDD ‘14. Model complexities of these works are discussed in other works [2]. Note that Deepwalk is one of the simpler models and can still perform similarly and sometimes better than other complex models. We have added text in blue in the conclusion for the same.
>
> New title: We have changed the title of our paper "Benchmarking and Analyzing Unsupervised Network Representation Learning and the Illusion of Progress".
>
> Description: We have added a description of the methods in the appendix for all the methods.
>
> Visual illustration of dataset preprocessing: We have added the two diagrams in the appendix section titled “Link Prediction Dataset Statistics”. These diagrams illustrate the dataset preprocessing procedures.
>
>
> Default hyperparameters: We do use the default hyperparameters specified in the original paper. Most of the papers, use the default parameter to report the results. For example, Grarep sets the parameters of Deepwalk as “window size as 10, walk length as 40, number of walks as 80” and does not tune its parameter. The default parameters of the methods are shared in the appendix titled “Default Parameters”. In H. Default vs Tuned Parameters, we compare the performance of methods with default vs tuned parameters.
>
>
> References
> 1. Huang, Zexi, Arlei Silva, and Ambuj Singh. "A Broader Picture of Random-walk Based Graph Embedding." Proceedings of the 27th ACM SIGKDD Conference on Knowledge Discovery & Data Mining. 2021.

---

> > ### Comment · Reviewer_tvky · 2022-05-23
> > **Updates**
> >
> > Thank you for addressing my concerns. However, some of my concerns still remain. Details are as follows:
> >
> > **Regarding the claim on benchmarking**: Yes, I acknowledge the authors' efforts in benchmarking various methods on several datasets, from which the authors observe that most of them are not far good than DeepWalk. However, the benchmarking is limited in terms of 1) few recent deep learning methods and datasets, and 2) only conducted on the performance of downstream tasks, not on the efficiencies and their actual behaviors.
> >
> > **Analyzing node properties**: Then, could Node2Vec capture node degree or clustering coefficient? I argued that downstream performance alone is not sufficient for benchmarking various methods, and there could be more metrics we can use and learn from them.
> >
> > **Hyperparameter tuning**: Thank you for your explanation on tuning. I now understand that the authors did not use validation sets, but just did tune the hyperparameters in the performance of test sets on 5-fold cross-validation, only for the classifier. However, my concern about the lack of validation data is not completely clear, while I'm not so confident that this is a conventional setup for the targeted task.
> >
> > **OGB datasets**: My intention was that, it would be highly valuable to benchmark various methods with large-scale and recent datasets from OGB. However, for me, the datasets in the paper also seem extensive, thus this point becomes not the major weakness, and I rather would like to just suggest to additionally use the larger and more recent benchmark datasets, if possible.
> >
> > **Methods**: Thank you for including the suggested method [1]. As the benchmarking is only performed on the graphs with node features (i.e., not suitable for [2, 3, 4]), I think it is better to tone down the claim, like this paper targets a certain type of node-featured graphs.
> >
> > **Model complexity**: Thank you for your clarification. I still believe including efficiencies and then comparing various methods under both their performances and efficiencies highly improve the paper, which is not clearly done so far in this paper, as well as most previous benchmark studies.
> >
> > **New title**: Thank you for reflecting my suggestion. The new title seems really good for me.
> >
> > **Description of methods; Illustration of data preprocessing; Description of default hyperparameters**: Thank you for comprehensively doing so. I really appreciate them.
> >
> > ---
> >
> > Overall again, I really appreciate the authors' efforts on benchmarking several methods, and showing the progress of unsupervised graph embeddings is still not far from DeepWalk. However, the claim with performance alone seems not perfectly sufficient for benchmarking several methods, for me.

---

> > > ### Author Response · Authors · 2022-05-28
> > > **Response on Updates**
> > >
> > > Comment on benchmarking: We have added a recent method published in KDD (Huang et al. August 2021). As noted by us earlier and also in the conclusions of our paper - scalability and  efficiency is not the focus of our paper.  We agree with the reviewer that scalability is an important consideration and certainly something both us and others plan to examine in the future (we have cited some initial ideas along these lines on this topic).  However to scope our paper we decided not to consider it in the current analysis, This is in line with several previous efforts  including (Perozzi et al., 2014; Grover & Leskovec, 2016; Cao et al., 2015; Qiu et al., 2018; Wang et al., 2017; Ou et al., 2016; Tang et al., 2015; Tsitsulin et al., 2018; Wang et al., 2016a; Kipf & Welling, 2016; Abu-El-Haija et al., 2018; Hamilton et al., 2017a; Velickovic et al. (2019); Huang et al. (2021). We are not entirely clear on what the reviewer means by “actual behavior”. We are showing the “actual behavior” of these methods on link prediction and node classification tasks.
> > >
> > >
> > > Comment on node properties: We don't disagree. In science, there is always the ability to learn more. However, a similar point  can be  raised on pretty much  any paper on machine learning  because there's always some aspect  one can learn more on! Please also see  our response to all the reviewers on May 15th.
> > >
> > > Comment on hyperparameter tuning: Our focus is on the “unsupervised” network representation learning methods.
> > > 1. We perform the grid search over the method parameters and get a score on the test split for all possible parameters in the grid. We report the best score from this search for each method.
> > > 2. If in lieu of the above we had used a traditional train, validate and test approach where we only used the validation set to tune hyperparameters (on the same grid)  it is quite likely that this set of parameters so chosen would result in sub-optimal test performance (it should be patently obvious that 1 is an upper bound on 2 - when we are searching on the same grid points).
> > > 3. We also note that such a strategy was previously employed by others (see Abu-El-Haija et al., 2018; Huang et al. 2021, Ou et al., 2016; and Wang et al., 2017.).  The use of validation set for link prediction is neither specified in the paper nor shared in the source code by Grover & Leskovec, 2016 - based on this we assume they employed a similar strategy as well.
> > >
> > >
> > >
> > >
> > > Comment on methods: We agree and have changed the title to reflect this. We have also added text specifying that we focus on graphs with no node features (See blue text at the end of Section 5).
> > >
> > >
> > > Comment on model complexity: Model complexity is not the focus of the paper. In general, model complexity has not been the focus in the network representation learning literature. We agree with the reviewer that it is important but again we refer back to our note of May 15th. Additionally, we will note that there is a lot of work in this literature where efficiency is not the focus. For instance, the following papers do not analyze efficiencies: Perozzi et al., 2014; Grover & Leskovec, 2016; Cao et al., 2015; Qiu et al., 2018; Wang et al., 2017; Ou et al., 2016; Tang et al., 2015; Wang et al., 2016a; Kipf & Welling, 2016; Abu-El-Haija et al., 2018; Hamilton et al., 2017a; Velickovic et al. (2019); Huang et al. (2021). As pointed out in the paper, the model complexity is studied in other studies (Moon et al. (2019); Liang et al. (2018a)).

---

> > > > ### Comment · Reviewer_tvky · 2022-05-31
> > > > **Final Updates**
> > > >
> > > > I understand that benchmarking unsupervised network representation methods in terms of scalability and efficiency is not the main focus of this work. However, the boundary -- evaluating existing methods only with node classification and link prediction performances -- limits the merits and significance of this work aiming at benchmarking.
> > > >
> > > > Regarding the comment on node properties, I was curious about why DeepWalk shows decent performance compared to existing methods. I believe that why DeepWalk yields good results should be analyzed further (for example, with analysis of capturing heuristics such as node properties which is my own suggestion), however, this is not shown in this work.
> > > >
> > > > Regarding hyperparameter, I acknowledge that the authors follow the existing experimental settings. However, I am not so confident that the mentioned hyperparameter tuning schemes are whether correct, and I believe the research community including this work can improve the hyperparameter setups.

---

### Review · Reviewer_X49W · 2022-04-27

**Summary Of Contributions:**

This paper surveys several recent methods for graph representation learning. The main contribution is a detailed set of experiments and analyses to evaluate whether progress has been made on the standard downstream tasks of edge prediction and node classification. This includes proposing a large collection of standard datasets and several simple baselines. The authors conclude that fairly little progress has been made in recent years, with classical baselines performing well and with little statistically significant improvements over DeepWalk (2014).

**Broader Impact Concerns:**

This paper's findings are agnostic to whether graph representations satisfy broader impact considerations such as fairness and interpretability. The authors should state this or perform additional experiments to strengthen their contributions. Analyzing the per-class error rates of node classifiers (or other more rigorous fairness metrics) in addition to micro/macro F1 would help determine whether any progress has been made in this direction.

**Requested Changes:**

### Critical
- Include more recent related work on network representations ([2-6] for a non-exhaustive sampling)
- Improve methodology for constructing train/test sets and tuning hyperparameters (see above)
- Merge discussion/results of [1] from the Appendix into the rest of the paper
- Show the impact of evaluation strategy, context embeddings, etc. on the ranking of representation methods for each task (see above)

### Non-critical
- Evaluation
    - Include Euclidean distance and mean average precision for link prediction
- Presentation
    - Include p-values in box plot figures (see above)
    - Dataset Specific Analysis subsections 7.1.G and 7.2.G could be summarized in a table of dataset property $\times$ Pearson correlation for each task
- Misc. Questions
    - Is the assumption $d << |V|$ fundamental, or is this due to computational limitations?
    - How is concatenation a "binary function on node embeddings"?
- Typos
    - "several drill down analysis" should be "analyses"
    - "lever a learning rate parameter" should be "leverage"


**Strengths And Weaknesses:**

In general I am evaluating this submission's claims based on the following questions:
- Are the methods studied recent/state-of-the-art?
- Are they evaluated objectively/thoroughly?
- Do the empirical findings support claims of underperformance?

The paper is strongest on the last question, i.e. insightful conclusions follow from the empirical results. However, the first 2 can be improved significantly

### Strengths
- Good example of a survey paper including extensive empirical study to confirm/deny/unify previous findings
- Includes simple baselines based on neighborhood overlap and classical node features
- Source code is publicly available
- Well written and organized

### Weaknesses
- Methods are not recent
    - All methods considered are from 2018 or earlier, which hurts the claim that they are recent and/or state-of-the-art. The most recent [1] appears only in the Appendix, and is 3 years old at the time of this review. Even some important circa-2018 lines of work such as hyperbolic embeddings [2,3] are missing. An updated version of the paper would include more recent work, for example [4-6] and references therein
- Concerns about evaluation/methodology
    - How standard is the construction of train and test sets? For example, "Nodes present in the test set, but not present in the training set, are deleted from the test set." How many nodes does this remove? Another approach would be to learn representations for these nodes during training using negative edges. This could contribute to why so many methods often perform very similarly in experiments. Therefore, it deserves a more detailed justification and/or comparison of multiple approaches.
    - The hyperparameter search [0.1, 0.5, 1.0] learning rate $\times$ [50, 100] epochs could be quite limiting, and could be improved with learning rate decay and/or early stopping
- Additional evaluation
    - Following [2-3], it may be beneficial to evaluate link prediction score with mean average precision in addition to AUC. Precision of positive edges emphasizes the representation's local structure
    - Euclidean distance is another common ranking score which could produce different performance than dot product ranking
    - Many experiments evaluate the impact of a system parameter by comparing the average AUC or F1-score improvement for each algorithm. These insights are usually consistent with intuition, for example nonlinear classifiers are more expressive and should perform better than linear classifiers. Additionally, the authors could show how the relative ranking of methods changes with evaluation strategy, context embeddings, etc.. Do the best methods get better, or do weaker methods overtake stronger ones?
    - Link prediction and node classification are important downstream tasks for graph representations. The paper would be strengthened by considering additional tasks such as graph classification (e.g. [7]), or by explaining why this is out of scope
- Presentation
    - Box plots are helpful but could be improved by adding vertical gridlines and the statistical significance discussed in the text

[1] Veličković et al., Deep Graph Infomax, ICLR 2019. https://arxiv.org/abs/1809.10341

[2] Nickel and Kiewla, Poincaré Embeddings for Learning Hierarchical Representations, NeurIPS 2017. https://arxiv.org/abs/1705.08039

[3] Nickel and Kiewla, Learning Continuous Hierarchies in the Lorentz Model of Hyperbolic Geometry, ICML 2018. https://arxiv.org/abs/1806.03417

[4] Gasteiger et al., Diffusion Improves Graph Learning. NeurIPS 2019. https://arxiv.org/abs/1911.05485

[5] Agarwal et al., Towards a Unified Framework for Fair and Stable Graph Representation Learning, UAI 2021. https://arxiv.org/abs/2102.13186

[6] Liu et al., Graph Self-Supervised Learning: A Survey. https://arxiv.org/abs/2103.00111

[7] Togninalli et al., Wasserstein Weisfeiler-Lehman Graph Kernels. NeurIPS 2019. https://arxiv.org/abs/1906.01277

---

> ### Author Response · Authors · 2022-05-15
> **Response from the authors**
>
> We thank the reviewer for the comments and valuable feedback. Please find below our detailed response.
>
> Recency of methods: There is a stream of publications that do not perform a careful analysis of baselines and we wanted to know if there are real scientific advances in the graph representation learning field. The conclusion of our study suggests that there are few methods that offer marginal improvements. In general, there has not been a large substantive advance since Deepwalk (which was published in KDD 2014). Please note that we perform a rigorous evaluation of the selected methods – a point supported by Reviewer 1. Yet, we have added experiments with a recent method [1] published in KDD 2021. We observe that even after the incorporation of the new method, our claim of the illusion of progress still holds. We are currently running experiments with Poincare [4] method but the results are not good as other methods. Please find below our rationale for why the following methods are not well suited for our evaluation task.
> 1. Reference 2 and 3 are proposed for hierarchical datasets and perform poorly on general datasets.
> 2. Reference 4 (GCC [2] method) introduces a generic two-step pre-processing that recommends the usage of a sparsified higher-order transition matrix (Page rank and Heat diffusion) instead of the original adjacency matrix. It is mainly aimed at substituting a simple transition matrix with a higher-order transition matrix for graph convolutions in GNNs. Despite that it can be used as the base transition matrix for any embedding methods, we decide not to compare this method as it is not a standalone embedding method and we have kept preprocessing methods out of the scope of this work.
> 3. Reference 5 is proposed for learning fair node representations and requires sensitive features.
> 4. Reference 6 contains most of the references to methods that operate on attributed graphs. Other mentioned references such as DeepWalk, Node2vec, and LINE are included in our baselines.
> 5. Reference 7 is proposed for the graph classification task. In our work, we focus on node classification and link prediction tasks.
>
> -----
> Poincare Results
>
> Node Classification Result with Logistic Regression with the train-test split of 50:50.
> 1. PPI : 0.06 Micro-f1
> 2. Wikipedia: 0.40 Micro-f1
> 3. Blogcatalog:  0.25 Micro-f1
>
> Link Prediction Result with Classifier
> 1. PPI:  0.627 AUROC
> 2. Wikipedia: 0.829 AUROC
> 3. Blogcatalog:  0.606 AUROC
>
> Deepwalk on the other hand performs extremely well as compared to Poincare [4] method on the above datasets.
>
> -----
>
> Construction of train and test sets: The evaluation methodology is for link prediction.  Here, we treat 20% of the edges of the graph as test edges. The graph formed with the rest 80% of the edges is treated as training graph. The train graph has around ~80\% edges while the test graph has 20\% edges. The statistics of the folds are shared in Section C titled (Link Prediction Dataset Statistics) and Table 4 and Table 5 in the appendix.  As we can see from these tables, for medium and large datasets, our test graphs have significant size. We have also added a figure that explains the creation of train/test splits.
>
> Hyperparameter search: We have initially explored various learning rates and epochs and found that in general, the selected hyperparameters work well.
>
> Mean Average Precision : We have added the Mean Average Precision Plot in Appendix (Section D titled “Link Prediction: Average Precision”).
>
> Euclidean distance: We are not sure why euclidean distance is being suggested. The dot product-based evaluation is commonly used by the existing literature (Ou et al. (2016); Kipf & Welling (2016); Abu-El-Haija et al. (2018); Wang et al. (2016a); Goyal & Ferrara (2018).). We observe that the classifier improves the link prediction performance of the methods.
>
>
>
> ---
> References
> 1. Huang, Zexi, Arlei Silva, and Ambuj Singh. "A Broader Picture of Random-walk Based Graph Embedding." Proceedings of the 27th ACM SIGKDD Conference on Knowledge Discovery & Data Mining. 2021.
> 2. Qiu, Jiezhong, et al. "Gcc: Graph contrastive coding for graph neural network pre-training." Proceedings of the 26th ACM SIGKDD International Conference on Knowledge Discovery & Data Mining. 2020.
> 3. Goyal, Palash, and Emilio Ferrara. "Graph embedding techniques, applications, and performance: A survey." Knowledge-Based Systems 151 (2018): 78-94.
> 4. Nickel and Kiewla, Poincaré Embeddings for Learning Hierarchical Representations, NeurIPS 2017. https://arxiv.org/abs/1705.08039

---

> > ### Author Response · Authors · 2022-05-15
> > **2nd Response from the authors**
> >
> > Ranking of methods: We have added a section for the ranking of methods based on the Link Prediction evaluation strategy. The results are added in the appendix titled “Link Prediction: Ranking”. We rank the methods based on their average AUROC score. If the average AUROC between the two methods is within 0.5 (*100) points, then they share the same ranks.  We observe that for random-walk-based methods such as Deepwalk, Node2vec, RW_framework as well as the family of methods represented by NetMF (LINE, Deepwalk, Node2vec), the use of classifier improves the link prediction performance. Moreover, even though the rank for VAG increases, the overall average AUROC of VAG still improves.  For the rest of the methods, the ranking is similar (rank difference <= 4) with respect to the evaluation strategy.
> >
> > Graph classification: Our selected methods learn node representations of the graph and our selected datasets are not suitable for the graph classification tasks. The graph classification task requires different datasets and a different set of methods.
> >
> > Assumption: The assumption of having embedding dimension d<<|V| is fundamental in the graph representation learning field.
> >
> > Concatenation operation takes two inputs (node embeddings of node u and v from the edge (u,v)) and outputs one result, hence it is a binary function.
> >
> > Vertical gridlines for box plots: Thanks for the suggestion. We have added gridlines to the boxplots.

---

> > > ### Comment · Reviewer_X49W · 2022-05-25
> > > **Update**
> > >
> > > Thank you for the thorough response, applying several suggested changes, and answering several clarification questions.
> > >
> > > - Scope: The initial paper's title and claims were stated so broadly that they were at risk of being misinterpreted as "Most of the past decade's scientific advances in network representation learning are not reproducible." Such a strong empirical claim that would have to meet a very (perhaps infeasibly) rigorous empirical standard. In the rebuttal, the the claims are much narrower in scope (consider moderately-sized datasets without node features or hierarchical structure, evaluate downstream node classification/link prediction accuracy). No set of experiments can be 100% exhaustive, which is why it's important to introduce all claims in the context of this particular combination of datasets, preprocessing, and evaluation.
> > >
> > > - Preprocessing: Thank you for adding the figure that describes train/test split. I am still not sure if this is standard and/or consistent with previous work. For example, one could sample test edges in a way that keeps the graph connected.
> > >
> > > - Thank you for adding additional methods and metrics which make the empirical evaluation more extensive. Social networks such as Reddit would be more reasonable evaluation for hierarchical methods, but the initial findings are important data points that strengthen the paper.
> > >
> > > - In addition to dot product, distance is also commonly used in the literature, for example mean average precision of Euclidean vs. Poincare embeddings (Nickel and Kiewla, 2017).

---

> > > > ### Author Response · Authors · 2022-05-28
> > > > **Response on Update**
> > > >
> > > > Comment on scope: We are not claiming that the author's work is not reproducible.  "Reproducibility is defined as obtaining consistent results using the same data and code as the original study" [1]. In most of the cases, our reported results are at least as good as the authors method, if not better. Hence the results are clearly reproducible. Our observation is that oftentimes when comparing against strawman solutions, researchers do not employ full tuning of strawman.  They often employ default parameters for strawman and many a time, the default parameters result in sub-optimal performance (See Figure 4 and Figure 14).
> > > >
> > > >
> > > > Comment on hierarchical method: We have almost compared the experimental analysis of PoinCare. We find it is  unable to handle moderately large datasets like Reddit on our machines. As shared earlier, all results we have obtained so far suggest that Poincare is performing strictly worse than Deepwalk on the datasets Poincare can execute on. We plan to include results on this method once all experiments finish.
> > > >
> > > > Additional Results on Poincare.
> > > >
> > > > Node classification results (Micro-f1) with Logistic Regression with the train-test split of 50:50.
> > > >
> > > > Microsoft: 0.35, Flickr: 0.17
> > > >
> > > > Link Prediction Result (AUROC) with Classifier
> > > >
> > > > Microsoft:0.838, Flickr: 0.825, WikiVote: 0.862, P2P-Gnutella: 0.691, Epinions: 0.750
> > > >
> > > >
> > > > References
> > > > 1. https://www.nationalacademies.org/news/2019/09/reproducibility-and-replicability-in-research

---

### Review · Reviewer_V52y · 2022-04-30

**Summary Of Contributions:**

This paper presents an extensive benchmarking effort comparing popular unsupervised graph representation learning methods, and concludes that much of the reported progress is illusory. Much work in the area (as also noted by the authors) seems to be motivated by the extreme simplicity of DeepWalk, which seems to tantalizingly suggest that development of methods that can improve up on it should be easy. As a consequence, many new methods have been proposed. However, as might be somewhat frustratingly clear to many practitioners working in the area, it has never been quite clear if any of these methods offer any significant benefit despite the reported results in papers. In many of these papers, context-specific/task specific baselines are rarely considered, and unfortunately the baselines are also set to default parameters and rarely tuned. This situation necessitates a careful benchmarking of these methods, a gap which this paper fills. Empirical work of a similar flavour was done by Goyal and Ferrera (2018) and Khosla et. al (2019), but the current paper could be considered more current and rigorous (amounting to a total of 375,000 experiments).

Some of the main contributions + findings of the paper include:
- Providing a standard evaluation protocol to allow fair comparison of such methods, including for the careful tuning of all the methods and the baselines considered.
- Benchmarking many task-specific baselines that have been proposed by the network community (and rarely considered in the DL literature), along with adding context specific information.
- When compared rigorously, most of the methods perform similarly to DeepWalk, with the exception of MNMF and WYS, which offer a slight advantage.
- Task-specific baselines can often be competitive with all the methods considered.
- Careful tuning of various baselines can lead to somewhat surprising gains (compared to reported in many papers), even for methods such as EigenMaps of Belkin and Niyogi.
- Adding context specific embeddings generally helps performance for methods such as LINE and WYS, while it hurts performance for the factorization methods.

Some secondary findings include:
- Datasets with high sparsity tend to have poor link prediction performance across the methods considered. Similarly, link prediction performance seems to be poor on datasets with a low clustering coefficient.
- Several node classification heuristics offer competitive performance to many recent methods when the number of classes are low.

**Broader Impact Concerns:**

No impact concerns.

**Requested Changes:**

I would prefer to see some more analysis of Deep Graph Infomax. It seems to perform poorly compared to DeepWalk, however the discussion has been relegated to the appendix and is carried out for a couple of cases. Preferably it should be treated as a first class citizen in the paper wherever possible and added along with other methods.  This is because it is indeed quite popular (and also does not quite give good performance). A brief discussion in the appendix (pointed at by "community feedback" in the main article) comes across as a little off.

**Strengths And Weaknesses:**

Strengths:
- Paper is written very clearly and is a pleasure to read.
- Valuable benchmarking and evaluation contributions .
- Consideration of task-specific baselines and heuristics and showing they too can be competitive with various DL methods.
- Providing evidence for what has been an open secret in the GRL community -- that most methods don't offer any significant advantage -- if all the baselines are tuned carefully.
- Some dataset level analysis to understand poor performance in some and good performance in others.

Weaknesses:
I could suggest a bunch of more recent baselines and also to evaluate heterogeneous methods such as metapath2vec. However, I am quite happy with the paper's contribution (and the methods considered) in its current state, so I would resist that temptation (except for one case -- see below).

---

> ### Author Response · Authors · 2022-05-15
> **Response from the authors**
>
> We thank the reviewer for their comments and valuable feedback.
>
> As suggested we have included Deep Graph Infomax (DGI) results in the main plots/figure, viz: Figures 1, 2, 8, 9, and 10. From the Illusion of Progress Figures 2 and 10 for the link prediction and node classification tasks, it is clear that DGI performs poorly in the featureless unsupervised node representation learning task.
>
> DGI learns local node representations that maximize its mutual information with a global graph feature. This is achieved with Noise Contrastive Loss which learns a discriminator that scores true global-local pairs higher in comparison to fake/noisy global-local pairs. While DGI is a popular unsupervised approach to learning node representations in graphs, it is ill-suited for the task in the absence of node features. This is mainly because the input data and target data are non-stationary in the DGI network. In our case, the target, the global summary is constantly changing along with neural network weights and the input embeddings. Whereas in graphs with features, only the neural network weights and the global summary is changing, that too not by much as it is only a mere average of all the node features. Additionally, in DGI there is no finer loss at a first-hop level to encode finer structural information like with classical link prediction loss of learning to detect edges. Due to these two reasons, DGI is ill-suited for the task of unsupervised node representation learning in the absence of features. As it performs poorly and is ill-suited for these tasks as seen in the primary results figure, we did not include DGI in other analyses.

---

### Author Response · Authors · 2022-05-15
**Response**

As the first reviewer points out that almost any evaluation of an ML method published in the literature can be questioned on parametric choices, additional method comparisons,  and results on additional datasets. If the reviewers wish us to run results on new datasets, new methods, or new parametric settings,  there should be a very clear rationale provided on why these additional sets of results are required -beyond the extensive study we have already reported. These experiments cost a lot of time and a lot of money (1000s of dollars) (given the exhaustive grid search on parameters, methods, and datasets). We have done our best to respond to some of these requests already.  As such we have provided a more extensive evaluation than pretty much anything we have seen in the literature in this space with a wide coverage of a wide range of methods, datasets, and commonly used parametric settings. Our evaluations are much more comprehensive than the previous efforts by Goyal and Ferrera (2018) and Khosla et. al (2019).  We also wish to point out that our choice of methods (popular, highly cited) do limit our ability to evaluate results broadly on very large datasets such as OGB - for example, the following methods: HOPE, SDNE, NetMF, MNMF, VAG, WYS, and DGI simply do not scale to very large datasets such as OGB. That said as noted previously many of the datasets we have evaluated already do share similar characteristics with the OGB data.  Also as we note in our paper (see Conclusions), scalability evaluation is out of the scope of our study.

---

### Decision · Action_Editors · 2022-06-01

**Recommendation:** Accept with minor revision

**Comment:**

This paper examines a number of existing approaches for unsupervised network representation learning and identifies under what conditions each method performs well. Using several different datasets, the paper shows that several recently proposed improvements provide marginal gains over strong baselines and points to the need for more rigorous tuning for baselines. The paper also analyzes the role of various factors of the task (e.g. neighborhood context) on the relative performance of the different algorithms.

The reviewers find the paper to be well-written, the study to be extensive in testing the findings of prior work and providing sufficient insights that will be of interest to the community, but also note some areas for improvement. Hence, I am recommending acceptance subject to the following revisions:
1. The authors should tone down the claims to make them more accurately reflect the empirical studies performed, including the scope of datasets, methods, preprocessing, downstream tasks (node classification, link prediction) investigated. This will entail some rewriting to parts of the paper and the addition of these caveats at places like the abstract, introduction and results.
2. Related to the above, the paper should provide more justification/discussion on why a method like DeepWalk serves as a good baseline and why some of the more recent methods mentioned by the reviewers are not considered. The authors mention some of this in their response, but this should be incorporated into the paper.
3. The paper should also clearly state the procedure for hyperparameter tuning, including the range of hyperparameters tried out, cross-validation method, etc. Also, clarify the preprocessing procedure and discuss how it compares to preprocessing in prior work. And for both these areas, discuss any limitations/caveats in the choices that may apply to the reader’s interpretation of the results.
4. Finish incorporating the promised results for the Poincare method which is currently in progress.

While not a required revision, I also highly encourage the authors to add some fine-grained analysis (e.g. error analysis) to shed some more light on why DeepWalk remains a strong baseline and to provide actionable insights for future work to improve these methods, which I believe will further strengthen this paper. The authors are in especially a good position to do this due to their extensive empirical study across several datasets that rules out some confounding factors usually arising in smaller-scale experimentation.

---

> ### Author Response · Authors · 2022-06-17
> **Camera-ready Version**
>
> We would like to thank the reviewers for providing helpful suggestions and comments. We have included all of the following changes to this final version:
> 1. Rewrote parts of the paper and narrowed down the claims (in abstract, introduction and experiments (section 7)).
> 2. Added justification for why DeepWalk serves as a good baseline (in section 7.1.A and section 7.2.A)  and rationale for why some of the more recent methods mentioned by the reviewers are not considered (in section 3).
> 3. Clarified preprocessing procedure and discussed how it compares to preprocessing in prior work (in section 6.1)
> 4. Added Poincare results (in section 7 and appendix).
> 5. We have expanded on the analysis in the latest version. We also have stability analysis (section 7.1.A and section 7.2.A and see Figure 7 and Figure 13).
>
> Hope this resolves all the concerns from the reviewers.